# Koopman Universal Neural Dynamic Operator: Achieving Fully Explicit Expression Identification for Nonlinear Dynamical Systems

## Abstract

Complex nonlinear systems permeate various scientific and engineering domains, presenting significant challenges in accurate modeling and analysis. This paper introduces the Koopman Universal Neural Dynamic Operator (KUNDO), a groundbreaking framework that bridges the gap between data-driven machine learning approaches and traditional mathematical modeling. KUNDO uniquely combines neural networks, Koopman operator theory, and the universal approximation theorem to achieve fully explicit expression identification for complex nonlinear systems. Our framework demonstrates remarkable efficiency in small sample scenarios, overcoming limitations of both classical physical models and black-box machine learning techniques. By learning Koopman-compatible basis functions through neural networks, KUNDO transforms strongly nonlinear dynamics into interpretable mathematical forms, greatly decreasing the limitations of human selection of basis functions without sacrificing predictive power. We present theoretical analyses of KUNDO's mathematical properties and validate its performance across diverse nonlinear systems. The results showcase KUNDO's potential to revolutionize system identification, offering new avenues for scientific discovery and engineering applications in fields such as climate science, financial modeling, and advanced robotics. This work presents a significant advance towards interpretable AI and data-driven modeling in systems analysis.

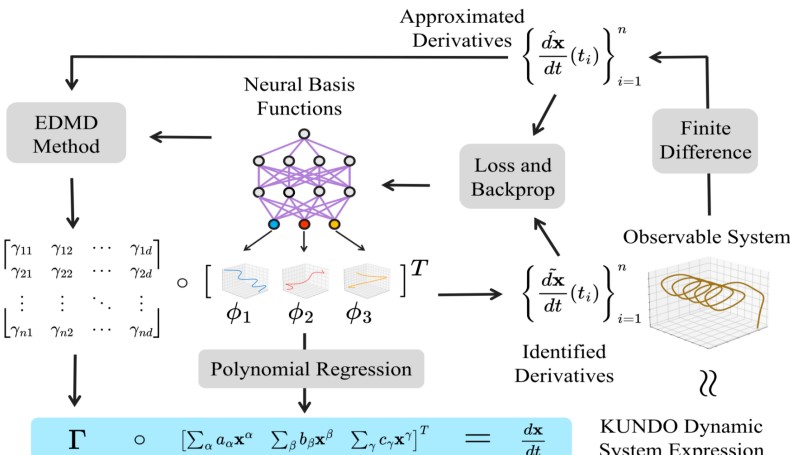

Figure 1: Schematic diagram of the KUNDO (Koopman Universal Neural Dynamical Observer) method for dynamical system identification and modeling. The system is optimized by minimizing the difference between $\{\frac{d\hat{\mathbf{x}}}{dt}(t_i)\}_{i=1}^n$ estimated from the observed $\mathbf{x}$ sequence using finite differences, and $\{\frac{d\tilde{\mathbf{x}}}{dt}(t_i)\}_{i=1}^n$ derived from the identified dynamical system.

## 1 INTRODUCTION

In contemporary science and engineering, accurately capturing the dynamic behavior of complex nonlinear systems while providing explicit mathematical expressions for them is an extremely challenging problem. Complex nonlinear systems are ubiquitous in various aspects of our lives, from climate change models to dynamic fluctuations in financial markets, from intricate interactions in biological systems to the control of advanced robotics. The behavior of these systems shapes our world, making the ability to understand and model their complex dynamics crucial for advancing fundamental science and solving practical engineering problems.

Traditional modeling methods often face numerous limitations when dealing with these complex nonlinear systems. Classical physical models and dynamical systems theory, while providing rigorous mathematical frameworks, struggle with strongly nonlinear systems. These methods typically rely on idealized assumptions, making it difficult to address the dynamic characteristics of complex real-world systems and thus limiting their applicability. Meanwhile, machine learning, especially deep learning techniques, has made significant progress in modeling complex phenomena in recent years. Through large-scale data, deep learning can capture the intricate dynamics of systems, demonstrating powerful predictive capabilities. However, these models are often viewed as "black boxes," lacking explicit mathematical explanations and interpretability. This "black box" nature not only diminishes the application value of the models in scientific and engineering fields but also hinders our in-depth understanding of the internal mechanisms of systems, impeding comprehensive analysis of system behavior.

The main challenge facing the field of modern system identification is: how can we design a method that captures the dynamic behavior of complex nonlinear systems while providing models with fully explicit mathematical expressions? Such a method must not only possess high predictive power but also achieve breakthroughs in interpretability and analyzability, especially for systems with higher dimensions and stronger nonlinearities.

This work proposes the Koopman Universal Neural Dynamic Operator (KUNDO) framework, aiming to address this challenge and provide a revolutionary modeling method for nonlinear systems. The core innovation of KUNDO lies in its ability to achieve fully explicit expression identification for complex nonlinear systems while demonstrating efficient performance in small sample scenarios. This breakthrough is achieved through the fusion of the powerful expressive capability of neural networks, the rigorous mathematical foundation of Koopman operator theory, and the universal approximation theorem.

The working principle of KUNDO is to use neural networks to learn a set of special basis functions that are compatible with Koopman operator theory. Leveraging the universal approximation theorem, these neural networks can approximate any continuous nonlinear function, enabling KUNDO to transform complex nonlinear dynamics into fully explicit mathematical expressions. This transformation preserves the essential characteristics of the system and provides unprecedented interpretability and analytical capabilities, especially for systems with higher dimensions and stronger nonlinearities. Moreover, KUNDO offers a new approach to understanding complex systems, capable of efficient learning and generalization even in scenarios with limited data.

The main contributions of this study include:

1. **Innovative Neural Network Architecture Design**: We propose a novel neural network architecture specifically designed to learn Koopman-compatible basis functions. This architecture can capture complex nonlinear dynamics while ensuring the learned expressions are interpretable.

2. **Integration with Koopman Operator Theory**: We develop a method that combines the learned basis functions with Koopman operator techniques to construct fully explicit system models. These models not only possess predictive capabilities but also provide mathematical analyzability for the system.

3. **Mathematical Property Analysis of the KUNDO Framework**: We conduct elaborate theoretical analyses of the mathematical properties of KUNDO, proving its effectiveness and robustness in handling complex nonlinear systems.

4. **Validation on Diverse Nonlinear Systems**: We validate the effectiveness of KUNDO on a series of challenging complex nonlinear systems, demonstrating its superiority in identifying fully explicit expressions and its efficient performance in small sample scenarios.

By integrating deep learning, dynamical systems theory, and the universal approximation theorem, KUNDO represents a new paradigm in machine learning. It not only enhances our understanding and predictive capabilities of complex nonlinear systems but also paves the way for interpretable AI and data-driven scientific discovery. We believe that this approach of combining machine learning with traditional scientific theories will play a crucial role in future AI systems, driving innovation in scientific discovery and engineering applications.

## 2 RELATED WORK

This section reviews relevant research in system identification and modeling for complex nonlinear systems, covering **traditional methods**, **Koopman operator theory** applications, **neural networks**, and **hybrid approaches**.

**Traditional methods** rely on physics-based mathematical models and classical dynamical systems theory (Ljung & Söderström, 1983; Ljung, 1998; Söderström & Stoica, 2002), including linear regression (Draper, 1998), autoregressive models (Box et al., 2015), and state-space representations (Durbin & Koopman, 2012). While effective for simpler systems, they struggle with high-dimensional, strongly nonlinear complex systems (Nelles & Nelles, 2020; Billings, 2013), often assuming linear or weakly nonlinear structures (Aguirre & Billings, 1995; Juang, 1994).

**Koopman operator theory** transforms nonlinear systems into linear operators in a high-dimensional space of observables (Koopman, 1931; Mezić, 2005). Dynamic Mode Decomposition (DMD) applies this theory to data-driven system identification (Schmid, 2010; Kutz et al., 2016). Extended methods like EDMD introduce nonlinear basis functions (Williams et al., 2015), while recent advancements include Hankel-DMD (Arbabi & Mezic, 2017) and time-delay embeddings (Brunton et al., 2017).

**Neural networks** have demonstrated remarkable capabilities in modeling nonlinear systems (LeCun et al., 2015), from RNNs (Elman, 1990) to LSTM (Hochreiter, 1997) and GRU (Cho, 2014). Recent innovations include Neural ODEs (Chen et al., 2018) and Graph Neural Networks (Scarselli et al., 2008; Battaglia et al., 2016; Gilmer et al., 2017). However, their "black-box" nature often limits interpretability (Schmidt & Lipson, 2009).

**Hybrid methods** combining Koopman theory and neural networks aim to leverage both approaches' strengths. Examples include DeepKoopman (Lusch et al., 2018), Koopman Autoencoders (Otto & Rowley, 2019), and EDMD with dictionary learning (Li et al., 2017). Despite improved modeling capabilities, these approaches face challenges in interpretability (Brunton et al., 2016a), stability for chaotic systems (Takeishi et al., 2017), and performance with limited data (Kaiser et al., 2021).

Recent research focuses on enhancing model **transparency and interpretability** in dynamical system modeling (Rudin, 2019). Physics-Informed Neural Networks (PINNs) incorporate physical constraints into neural network training (Raissi et al., 2019), while other approaches explore interpretable structures or regularization terms (Adadi & Berrada, 2018). However, a systematic framework deeply integrating traditional physical theories with machine learning methods is still lacking.

## 3 FRAMEWORK: KOOPMAN UNIVERSAL NEURAL DYNAMIC OPERATOR

This section provides a detailed description of the methodology of the Neural Network KUNDO (Koopman Universal Neural Dynamic Operator). KUNDO integrates Koopman operator theory, the universal approximation theorem, and deep learning techniques to achieve efficient modeling and prediction of complex nonlinear dynamical systems through two-layer Koopman-like mappings and system identification.

## 3.1 METHODOLOGICAL FRAMEWORK

The KUNDO method begins by concatenating the original state vector $\mathbf{x} \in \mathbb{R}^n$ and control input $\mathbf{u} \in \mathbb{R}^m$ to form the input vector $\mathbf{x}_u = [\mathbf{x}, \mathbf{u}] \in \mathbb{R}^{n+m}$.

A neural network $\Phi \colon \mathbb{R}^{n+m} \to \mathbb{R}^d$ is designed such that each neuron in the output layer corresponds to a distinct nonlinear basis function, explicitly outputting a single basis function value:

$$\Phi(\mathbf{x}, \mathbf{u}) = [\phi_1(\mathbf{x}, \mathbf{u}), \phi_2(\mathbf{x}, \mathbf{u}), \dots, \phi_d(\mathbf{x}, \mathbf{u})]^T, \tag{1}$$

where $d$ is the number of basis functions. This network can encompass various architectures, such as feedforward neural networks, convolutional neural networks (CNN), recurrent neural networks (RNN), graph neural networks (GNN), or combinations thereof.

Using these learned basis functions, a system dynamics model is constructed through a linear mapping akin to the Extended Dynamic Mode Decomposition (EDMD) framework:

$$\dot{\mathbf{x}} = \Gamma \Phi(\mathbf{x}, \mathbf{u}), \tag{2}$$

where $\Gamma \in \mathbb{R}^{n \times d}$ is the parameter matrix to be estimated. Given time series data $(\mathbf{x}_k, \mathbf{u}_k, \dot{\mathbf{x}}_k)_{k=1}^N$, the parameter matrix $\Gamma$ is estimated using the least squares method inspired by EDMD:

$$\Gamma = \arg\min_{\Gamma} \sum_{k=1}^{N} \|\dot{\mathbf{x}}_k - \Gamma \Phi(\mathbf{x}_k, \mathbf{u}_k)\|_2^2. \tag{3}$$

This formulation aligns with the EDMD approach, where the objective is to find the optimal linear approximation in the lifted space defined by the basis functions.

The optimization process involves iterative methods such as the Adam optimizer to simultaneously refine the neural network parameters and $\Gamma$, minimizing the overall error through backpropagation. Additionally, a closed-form solution leveraging the Moore-Penrose pseudoinverse can be utilized:

$$\Gamma = \dot{X} \Phi^+, \tag{4}$$

where

$$\dot{X} = [\dot{\mathbf{x}}_1, \dot{\mathbf{x}}_2, \cdots, \dot{\mathbf{x}}_N]^T, \quad \Phi = [\Phi(\mathbf{x}_1, \mathbf{u}_1), \Phi(\mathbf{x}_2, \mathbf{u}_2), \cdots, \Phi(\mathbf{x}_N, \mathbf{u}_N)]^T,$$

and $\Phi^+$ denotes the Moore-Penrose pseudoinverse of $\Phi$.

## 3.2 SYSTEM INTERPRETATION AND EXPLICIT EXPRESSION

To enhance interpretability and achieve explicit system identification, we approximate the KUNDO model's basis functions using **polynomial regression**. For each basis function $\phi_i(\mathbf{x}, \mathbf{u})$, we obtain:

$$\phi_i(\mathbf{x}, \mathbf{u}) \approx a_{i0} + \sum_{j=1}^{n} \sum_{p=1}^{P} a_{ijp} x_j^p + \sum_{k=1}^{m} \sum_{q=1}^{Q} b_{ikq} u_k^q + \sum_{j,k} c_{ijk} x_j u_k + \sum_{j,l} d_{ijl} x_j x_l + \sum_{k,l} e_{ikl} u_k u_l, \tag{5}$$

where $x_j$ and $u_k$ are state and control variables, respectively. The coefficients $a$, $b$, $c$, $d$, and $e$ are determined through regression. $P$ and $Q$ represent the maximum polynomial degrees for state and control variables.

Through this method, the dynamics of the entire system can be represented explicitly as

$$x_{j,t+1} = f_j(\mathbf{x}t, \mathbf{u}t) \approx cj0 + \sum k = 1^d c_{jk} \phi_k(\mathbf{x}_t, \mathbf{u}t), \quad j \in 1, \dots, n, \tag{6}$$

where $c_{jk}$ are coefficients derived from the parameter matrix $\Gamma$ through a linear transformation of its rows. Each variable and coefficient is clearly defined to maintain consistency with the overall dynamical system equations.

This explicit representation enhances the model's interpretability and facilitates various analytical tasks such as stability analysis and control design. While polynomial regression is used here, other methods like **Fourier series expansion** could also be employed, depending on the system's characteristics. The choice of method and the selection of maximum polynomial degrees $P$ and $Q$ involve trade-offs between approximation accuracy, computational efficiency, and interpretability. Balancing these factors is crucial to capture the essential dynamics without overfitting, ensuring an accurate and interpretable representation of the system.

## 3.3 System Prediction

Based on the identified explicit system model, we construct a continuous-time dynamic equation:

$$\dot{\mathbf{x}} = F(\mathbf{x}, \mathbf{u}) = \Gamma\Phi(\mathbf{x}, \mathbf{u}), \tag{7}$$

where $\Gamma$ is the estimated parameter matrix, and $\Phi(\mathbf{x}, \mathbf{u})$ is a vector composed of polynomial-approximated basis functions.

Using this model, we can predict the future state of the system. Given an initial state $\mathbf{x}_0$ and a series of control inputs $\{\mathbf{u}(t)\}_{t=0}^{T-1}$, we simulate the system trajectory using numerical integration methods, such as the Euler method:

$$\mathbf{x}(t + \Delta t) = \mathbf{x}(t) + F(\mathbf{x}(t), \mathbf{u}(t))\Delta t. \tag{8}$$

Through recursive calculations, we obtain state predictions for the system at future time points. This method combines the advantages of data-driven modeling and analytical expression, achieving an in-depth understanding and effective prediction of nonlinear systems. It represents an important development in modern system identification and prediction.

## 3.4 Theoretical Analysis

The KUNDO method introduces a dynamic embedding space defined by the neural network $\Phi(\cdot)$, achieving adaptive learning of state representation. This process can be viewed as a finite-dimensional approximation of the generalized Koopman operator:

$$\Phi(\mathbf{f}(\mathbf{x}, \mathbf{u})) \approx \Gamma\Phi(\mathbf{x}, \mathbf{u}), \tag{9}$$

where $\Gamma$ captures the evolution of the system in the embedding space. This approach not only extends traditional Koopman theory but also aligns with the EDMD framework, providing a new perspective for **spectral analysis of nonlinear dynamical systems**. Through the learned embedding and dynamic parameters, we can analyze **characteristic structures of the system, such as invariant subspaces and periodic orbits**, offering new tools for the qualitative analysis of complex systems.

Although neural networks may map states to high-dimensional spaces, the entire KUNDO framework acts as a form of implicit regularization. By learning an effective embedding, the method automatically identifies and retains the most relevant dynamic features, achieving data-driven dimension reduction. This ensures powerful modeling capabilities for complex nonlinear systems while controlling model complexity through the parameter matrix $\Gamma$, effectively preventing overfitting.

Overall, the KUNDO method represents a modern approach that combines theoretical guidance with data-driven techniques, opening new possibilities for the analysis and prediction of complex dynamical systems. It provides powerful tools to understand and interpret the intrinsic dynamic structures of these systems. More detailed analysis and mathematical properties can be found in Appendix A.1.

## 4 Experiments

This section systematically evaluates the effectiveness and superiority of the proposed Koopman Universal Neural Dynamical Operator (KUNDO) method through multiple complex systems with practical physical significance and engineering application backgrounds. We selected four typical nonlinear dynamical systems as experimental subjects, including generated data and real machine-collected data. We conducted detailed comparisons between KUNDO and various mainstream baseline methods to verify its effectiveness in various scenarios.

### 4.1 Experimental Subjects and Their Mathematical Modeling

To fully demonstrate the generalization ability of the KUNDO method and its applicability in systems of different complexities, we selected the following four representative nonlinear dynamical systems, generating trajectories under 100 different initial conditions to form the datasets:

### 4.1.1 TASK A: NONLINEAR SYSTEM

Task A deals with a parameterized nonlinear system (Strogatz, 2018) with the following dynamic equations:

$$\begin{cases} \dot{x} = a \cdot x, \\ \dot{y} = b \cdot y, \\ \dot{z} = c \cdot z + x \cdot y. \end{cases} \tag{10}$$

The system state vector is $\mathbf{x} = [x, y, z]^T$, with parameters set to three different cases(Perko, 2013): (1) $a = 1$, $b = -1$, $c = -1$, corresponding to a Saddle Point in phase space; (2) $a = -1$, $b = -1$, $c = -1$, corresponding to a Sink in phase space; and (3) $a = 1$, $b = 1$, $c = 1$, corresponding to a Source in phase space.

The simulation time range is $t \in [0, 20]$, using the fourth-order Runge-Kutta method (Press, 2007) for numerical integration, with 5000 sampling points.

### 4.1.2 TASK B: LORENZ SYSTEM

The Lorenz system is a classic chaotic system, known for its sensitivity to initial conditions and complex dynamic behavior (Lorenz, 1963; Tucker, 2002). Its dynamic equations are defined as

$$\begin{cases} \dot{x} = \sigma(y - x), \\ \dot{y} = x(\rho - z) - y, \\ \dot{z} = xy - \beta z. \end{cases} \tag{11}$$

The parameters are set to $\sigma = 10$, $\rho = 28$, $\beta = \frac{8}{3}$. The simulation time range is $t \in [0, 30]$, using the fourth-order Runge-Kutta method for numerical integration, with 10000 sampling points.

### 4.1.3 TASK C: REAL ROBOTIC ARM DATASET

To evaluate KUNDO's application capability in real engineering data, we used a self-collected robotic arm dataset on Flexiv Rizon. This dataset includes time series information of joint angles under different initial poses (joint configurations). Given target end-effector positions, inverse kinematics solution algorithms and the **Covariant Hamiltonian Optimization for Motion Planning (CHOMP)** algorithm (Zucker et al., 2013) from automatic planning algorithms were used to generate paths, which are used as the dataset. The generated paths were adjusted to the same length through interpolation methods, with each trajectory containing 1000 time steps. The data preprocessing techniques used include denoising and normalization.

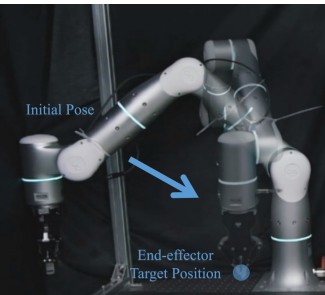

Figure 2: Robotic arm setup for motion planning dynamics dataset: illustration of initial pose and end-effector target position.

### 4.1.4 TASK D: ONE-DIMENSIONAL WAVE EQUATION

Task D involves a one-dimensional wave equation (Strauss, 2007; French, 2017) to evaluate KUNDO's performance in handling partial differential equation systems. The form of the wave equation is:

$$\frac{\partial^2 u}{\partial t^2} = c^2 \frac{\partial^2 u}{\partial x^2}, \tag{12}$$

where $u(\mathbf{x}, t)$ is the wave quantity, $\mathbf{x}$ is the spatial coordinate vector, and $c$ is the wave speed. For numerical solution using the finite difference method (LeVeque, 2007), the continuous spatial domain is discretized into a one-dimensional array of points. The simulation covers a time range of $t \in [0, 50]$ and a spatial range of $x \in [0, 10]$, with discretization steps $\Delta t = 0.01$ and $\Delta x = 0.1$. Periodic boundary conditions are applied to simulate an infinite domain (Fornberg, 1998). This

discretization results in a spatial vector $\mathbf{x}$ with 31 points. At each time step, the wave quantity $u$ is computed for each point in this spatial vector, generating a dataset with 5000 temporal sampling points. Each of these temporal samples contains the wave quantity values across all 31 spatial points, effectively creating a $5000 \times 31$ matrix of wave quantity values.

## 4.2 Selection of Baseline Methods and Rationale

To ensure fairness and comprehensiveness of comparison, we selected multiple representative baseline methods in the field of system identification. These methods include **SINDy**(Brunton et al., 2016b), **Latent Neural ODE** (Chen et al., 2018), **LSTM**(Hochreiter, 1997), **NTM**(Graves, 2014), **GPR**(Williams & Rasmussen, 2006). These methods cover a range from sparse models to deep learning, from linearization methods to memory-based models, as well as the ability to handle noise and uncertainty.

**SINDy** identifies explicit dynamic equations of the system through sparse regression, suitable for systems with clear physical mechanisms and offering good interpretability. **Latent Neural ODE (NODE)** as an implicit model can effectively capture complex nonlinear relationships but lacks interpretability. **LSTM** has strong expressive power, suitable for processing time series data, but similarly lacks physical interpretation of the model. **NTM** combines the capabilities of neural networks and Turing machines, possessing memory and complex computational characteristics, suitable for handling high-complexity dynamic systems. **Gaussian Process Regression (GPR)**, based on Bayesian theory, can provide uncertainty estimates for predictions, suitable for small sample and noisy data environments, with advantages in uncertainty assessment.

## 4.3 Experimental Design and Implementation

### 4.3.1 Model Training and Evaluation

For each system, we train **KUNDO** and each baseline method separately. KUNDO's neural basis function component is implemented using the Neural ODE framework, where the initial step involves mapping the input vector to a latent space with dimensions equal to the number of basis functions.

Other baseline methods also adopt their respective suitable training steps. The **SINDy** method includes two main steps: feature library construction and sparse regression; **Latent Neural ODE** involves latent space neural network design and integration of ODE solvers; **LSTM** and **NTM** include network structure design and optimization; **GPR** includes kernel function selection and Bayesian model training.

To ensure fairness in comparison, we strive to maintain consistency in the model complexity (such as number of network layers and parameters) across methods, avoiding performance bias due to differences in model capacity. For KUNDO, the default settings are a two-layer neural network with 256 neurons per layer, learning rate of $1 \times 10^{-3}$, 1000 epochs, and 11 basis functions.

### 4.3.2 Experimental Tasks and Evaluation Metrics

We designed five experiments to comprehensively evaluate the performance of each method. These experiments are as follows: **System Identification and Modeling Accuracy**, where the dataset is divided into training set (70%) and test set (30%), evaluating the similarity between predicted trajectories and actual trajectories using Mean Squared Error (MSE), Mean Absolute Percentage Error (MAPE), and Directional Accuracy (DA) as metrics; **Extrapolation Generalization Ability**, examining the prediction accuracy in unknown time periods by extending the test time range to 1.5 times the original (e.g., training in $t \in [0, 30]$, testing in $t \in [30, 45]$); **Noise Resistance**, evaluating robustness in noisy environments by adding Gaussian noise of different intensities (standard deviation $\sigma = 0.1$, $\sigma = 0.5$, $\sigma = 1.0$); **Small Sample Learning Experiment**, assessing performance with reduced training samples (10%, 20%, 30%, 40%, 50% of the original training set) for testing few-shot learning ability of methods; and **Parameter Sensitivity**, conducted on KUNDO using Task B, evaluating its sensitivity to hyperparameters including learning rate, number of neurons, and number of basis functions, using a fixed two-layer neural network structure across 1000 epochs of training. This comprehensive set of experiments aims to assess each method's modeling ability, generaliza-

tion, robustness to noise, performance under limited data, and in KUNDO's case, its response to different parameter settings.

## 4.4 EXPERIMENTAL RESULTS AND ANALYSIS



Figure 3: Comparison of phase space trajectories for the nonlinear dynamical system exhibiting saddle point behavior in Task A. From left to right: ground truth, KUNDO prediction, and Latent Neural ODE prediction.

### 4.4.1 SYSTEM IDENTIFICATION AND MODELING ACCURACY

In terms of system identification accuracy, KUNDO performed excellently in all experimental tasks. The specific results are shown in Table 1.

Table 1: Identification performance indicators of each method on different experimental tasks. "-" indicates numerical explosion in that task, unable to obtain valid results.

| Task | | Method | | | | | |
|---|---|---|---|---|---|---|---|
| | | KUNDO | LSTM | NODE | GPR | SINDy | NTM |
| Task A (Saddle) | MSE | **0.010($\pm$0.002)** | 0.012($\pm$0.002) | 0.035($\pm$0.005) | 0.038($\pm$0.005) | 0.041($\pm$0.006) | 0.028($\pm$0.004) |
| | MAPE (%) | **2.5($\pm$0.3)** | 2.7($\pm$0.3) | 8.9($\pm$1.0) | 9.5($\pm$1.1) | 10.2($\pm$1.2) | 7.0($\pm$0.8) |
| | DA (%) | **95($\pm$2)** | 94($\pm$2) | 90($\pm$2) | 88($\pm$2) | 85($\pm$2) | 91($\pm$2) |
| Task A (Sink) | MSE | **0.011($\pm$0.002)** | 0.013($\pm$0.002) | 0.037($\pm$0.005) | 0.040($\pm$0.005) | 0.043($\pm$0.006) | - |
| | MAPE (%) | **2.7($\pm$0.3)** | 3.0($\pm$0.3) | 9.3($\pm$1.0) | 10.0($\pm$1.1) | 10.7($\pm$1.2) | - |
| | DA (%) | **94($\pm$2)** | 93($\pm$2) | 89($\pm$2) | 86($\pm$2) | 84($\pm$2) | - |
| Task A (Source) | MSE | 0.029($\pm$0.004) | **0.021($\pm$0.003)** | 0.043($\pm$0.006) | 0.046($\pm$0.006) | 0.049($\pm$0.007) | 0.041($\pm$0.005) |
| | MAPE (%) | 3.3($\pm$0.4) | **3.2($\pm$0.4)** | 10.3($\pm$1.2) | 10.0($\pm$1.1) | 11.7($\pm$1.3) | 7.2($\pm$0.8) |
| | DA (%) | **91($\pm$2)** | 90($\pm$2) | 88($\pm$2) | 86($\pm$2) | 81($\pm$2) | 89($\pm$2) |
| Task B | MSE | **0.035($\pm$0.004)** | 0.038($\pm$0.005) | 0.065($\pm$0.008) | 0.069($\pm$0.008) | 0.072($\pm$0.009) | 0.062($\pm$0.007) |
| | MAPE (%) | **4.8($\pm$0.5)** | 5.5($\pm$0.6) | 12.2($\pm$1.4) | 13.2($\pm$1.5) | 14.0($\pm$1.6) | 11.5($\pm$1.3) |
| | DA (%) | **87($\pm$2)** | 83($\pm$2) | 81($\pm$2) | 79($\pm$2) | 71($\pm$2) | 82($\pm$2) |
| Task C | MSE | 0.020($\pm$0.003) | **0.019($\pm$0.003)** | 0.055($\pm$0.007) | 0.059($\pm$0.007) | 0.062($\pm$0.008) | - |
| | MAPE (%) | 4.5($\pm$0.5) | **4.3($\pm$0.5)** | 13.7($\pm$1.5) | 14.7($\pm$1.6) | 15.5($\pm$1.7) | - |
| | DA (%) | **92($\pm$2)** | 90($\pm$2) | 83($\pm$2) | 80($\pm$2) | 78($\pm$2) | - |
| Task D | MSE | **0.011($\pm$0.002)** | 0.015($\pm$0.002) | 0.025($\pm$0.003) | 0.029($\pm$0.004) | 0.031($\pm$0.004) | 0.016($\pm$0.002) |
| | MAPE (%) | **2.9($\pm$0.3)** | 3.1($\pm$0.3) | 6.2($\pm$0.7) | 7.2($\pm$0.8) | 7.7($\pm$0.9) | 4.0($\pm$0.5) |
| | DA (%) | **95($\pm$2)** | 94($\pm$2) | 90($\pm$2) | 89($\pm$2) | 87($\pm$2) | 93($\pm$2) |

Table 1 demonstrates KUNDO's strong performance across tasks, often surpassing baseline methods in MSE and MAPE. Notably, KUNDO achieves the best directional accuracy in all tasks, a critical metric for dynamical systems. While LSTM occasionally shows marginally better results in some metrics, KUNDO maintains competitive performance throughout. Figure 3 further illustrates KUNDO's superior ability to capture saddle characteristics compared to NODE in Task A. Crucially, KUNDO offers enhanced interpretability over LSTM, a significant advantage in analyzing complex dynamical systems.

### 4.4.2 EXTRAPOLATION GENERALIZATION ABILITY

To evaluate the model's predictive capability beyond the training time range, we conducted extrapolation experiments. These experiments were performed on the Lorenz system (Task B) and the Wave Equation (Task D), with results presented in Table 2 and Figure 4.

Figure 4: Extrapolation predictions: (a) Lorenz System, (b) Wave Equation. KUNDO vs LSTM in extended time range.

Table 2: Performance of Task B (Lorenz System) and Task D (Wave Equation) extrapolation prediction

| Method | Metric | Task B | Task D |
|--------|--------|--------|--------|
| KUNDO | MSE | **0.035** | **0.011** |
| | MAPE (%) | **4.8** | **2.9** |
| | DA (%) | **87** | **95** |
| LSTM | MSE | 0.038 | 0.030 |
| | MAPE (%) | 5.5 | 3.9 |
| | DA (%) | 83 | 81 |
| NODE | MSE | 0.065 | 0.045 |
| | MAPE (%) | 12.2 | 7.1 |
| | DA (%) | 81 | 75 |
| GPR | MSE | 0.069 | 0.049 |
| | MAPE (%) | 13.2 | 7.6 |
| | DA (%) | 79 | 79 |
| SINDy | MSE | 0.072 | 0.041 |
| | MAPE (%) | 14.0 | 8.7 |
| | DA (%) | 71 | 67 |

As evidenced by Table 2, KUNDO consistently outperforms other methods in extrapolation prediction across both tasks. For the Lorenz system (Task B) and the Wave Equation (Task D), KUNDO demonstrates significantly lower error rates and higher Direction Accuracy compared to baseline methods. This superior performance is particularly noteworthy given the complex, nonlinear nature of these systems. Figure 4 provides visual confirmation of KUNDO's capabilities, illustrating its ability to more accurately capture the intricate dynamics of both systems during extrapolation when compared to LSTM predictions. These results suggesting its potential for reliable long-term predictions in complex dynamical systems beyond the training range.

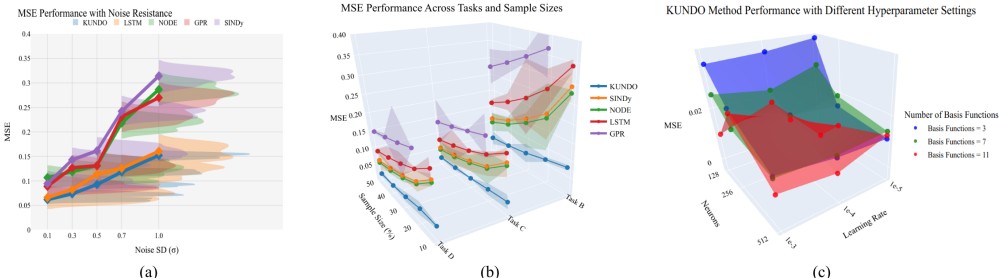

Figure 5: (a) MSE comparison of methods for Task A's sink case under varying noise. Lines show mean MSE; shaded areas indicate MSE distribution. (b) MSE performance comparison of different methods across three tasks (B, C, and D) and varying sample sizes. Shaded areas indicate error bounds. (c) 3D surface visualization of KUNDO method MSE performance across learning rates, neuron counts, and basis function numbers.

### 4.4.3 NOISE RESISTANCE

To evaluate the robustness of each method in noisy environments, we focused on the sink case of Task A, which represents a nonlinear system with an attractor. We added Gaussian noise of different magnitudes to this data. The results are visually represented in Fig. 5(a), with detailed numerical results provided by Table 3 in Appendix A.3.

Fig. 5(a) illustrates KUNDO's superior performance in the sink case of Task A across all noise levels. KUNDO consistently achieves lower Mean Squared Error (MSE) values with narrower distributions compared to other methods, indicating both better prediction accuracy and stability. This advantage is particularly evident at higher noise levels ($\sigma = 0.7$ and $\sigma = 1.0$), where KUNDO maintains low MSE with minimal spread while other methods' performance deteriorates. These results demonstrate KUNDO's robust noise resistance in capturing nonlinear dynamics with attractors, making it well-suited for real-world applications involving noisy complex systems.

### 4.4.4 SMALL SAMPLE LEARNING EXPERIMENT

We evaluated KUNDO's performance in small sample scenarios, comparing it with SINDy, NODE, LSTM, and GPR across Tasks B, C, and D. The experiment used 10% to 50% of the full training set (70 trajectories) in 10% increments.

As shown in Fig. 5 (b), KUNDO consistently outperformed other methods across all sample sizes and tasks. It achieved the lowest MSE values, with its advantage most pronounced at 10% sample size. All methods improved with increasing samples, but KUNDO maintained its leading role. LSTM showed the poorest performance, especially with small samples. SINDy and NODE performed better than LSTM, while GPR improved upon SINDy but still lagged behind KUNDO. More detailed numerical results are provided by Table 5 in Appendix A.3.

Notably, KUNDO exhibited low-performance variation in repeated experiments, further demonstrating its stability and reliability. These results not only showcase KUNDO's superior ability to perform accurate system identification with limited data but also highlight its potential for real-world applications where data scarcity is common.

### 4.4.5 PARAMETER SENSITIVITY

We investigated KUNDO's sensitivity to hyperparameters using Task B. Fig. 5(c) illustrates the impact of learning rate, neuron count, and basis function number on performance. Detailed numerical results are provided in Table 4 in Appendix A.3.

All tested learning rates (from $1 \times 10^{-5}$ to $1 \times 10^{-3}$) converged to similar performance levels. Increasing neurons from 128 to 512 generally improved accuracy. The optimal number of basis functions varied, with 7 often performing best. These findings suggest that while KUNDO is robust across a range of hyperparameters, fine-tuning can still yield marginal improvements in performance for specific tasks.

## 5 CONCLUSION, LIMITATIONS, AND FUTURE WORK

KUNDO uniquely combines Koopman operator theory with neural networks, achieving fully explicit expression identification for complex nonlinear systems. It outperforms mainstream methods in accuracy, extrapolation, noise robustness, and small sample learning. By learning Koopman-compatible basis functions, KUNDO transforms nonlinear dynamics into interpretable forms, reducing reliance on human expertise without sacrificing predictive power. The method demonstrates stable performance across various hyperparameter settings, showcasing its robustness and tunability. However, KUNDO's integration of EDMD and other optimization techniques increases computational complexity, resulting in longer training times compared to simpler models. Current implementation may not fully utilize GPU parallelization, potentially limiting scalability for large-scale or real-time applications.

Future work will focus on optimizing computational efficiency, particularly GPU utilization, to enhance KUNDO's applicability in real-time and large-scale systems. Exploring its potential in higher-dimensional systems and more complex dynamical tasks remains crucial. These developments will further establish KUNDO's role in advancing interpretable AI and data-driven modeling across scientific and engineering domains.

## REPRODUCIBILITY STATEMENT

We have implemented comprehensive and rigorous measures to ensure the reproducibility of our work. All experimental procedures, including data generation, model architecture, training protocols, and evaluation metrics, are thoroughly described in Section 4 of the main paper. To facilitate replication, we have open-sourced the complete implementation of our KUNDO (Koopman Universal Neural Dynamical Operator) method, which is available at the following anonymous repository:

https://anonymous.4open.science/r/kundo-BDBC/

Our experiments were conducted on a machine equipped with an NVIDIA GeForce RTX 3070 GPU and CUDA version 11.7, utilizing an Intel Core i7-10700 CPU @ 2.90GHz with 32GB RAM. All library dependencies, including PyTorch (version 1.9.0), TorchDiffEq (version 0.2.2), and Scikit-learn (version 0.24.2), are detailed in the environment configuration files within the repository.

To ensure complete reproducibility, we provide synthetic data generation code in our repository. These codes are meticulously designed to simulate a variety of complex dynamical systems, ranging from simple linear systems to highly nonlinear chaotic systems. We have also included scripts for data preprocessing and augmentation to ensure the quality and consistency of input data.

In the main text of our paper, we discuss in detail the critical hyperparameters that influence the performance of KUNDO. These include the number of layers in the encoder and decoder networks, the dimension of the latent Koopman space, and the learning rate.

For a comprehensive description of the KUNDO algorithm, including forward propagation, loss calculation, and backpropagation, the reader is referred to Appendix A.4. This appendix provides a step-by-step breakdown of the algorithm, ensuring that readers can fully understand and implement our method. We have also included detailed pseudocode in the appendix, as well as mathematical derivations of key functions, which may help readers gain a deeper understanding of the internal workings of the algorithm.

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

# A   APPENDIX

## A.1   THEORETICAL FOUNDATIONS AND MATHEMATICAL PROPERTIES

By combining Koopman operator theory, the universal approximation theorem, and deep learning techniques, the KUNDO method demonstrates significant advantages in modeling and predicting complex nonlinear dynamical systems. This chapter will provide detailed mathematical analysis and argumentation from aspects such as Koopman operator approximation, geometric properties of embedding space, convergence of parameter estimation, and the model's generalization ability and stability.

### A.1.1   DYNAMIC EMBEDDING SPACE AND FINITE-DIMENSIONAL APPROXIMATION OF GENERALIZED KOOPMAN OPERATOR

The KUNDO method defines a dynamic embedding space through a neural network $\Phi(\cdot)$, mapping nonlinear dynamical systems to high-dimensional linear spaces, thereby achieving a finite-dimensional approximation of the generalized Koopman operator.

**Nonlinear Dynamical Systems and Koopman Operator**    Consider a discrete-time nonlinear dynamical system described in state space as:

$$\mathbf{x}_{t+1} = \mathbf{f}(\mathbf{x}_t, \mathbf{u}_t), \tag{13}$$

where $\mathbf{x}_t \in \mathbb{R}^n$ is the system state at time $t$, $\mathbf{u}_t \in \mathbb{R}^m$ is the control input, and $\mathbf{f} \colon \mathbb{R}^{n+m} \to \mathbb{R}^n$ is a nonlinear mapping.

**Definition (Koopman Operator)** For any observation function $\phi \colon \mathbb{R}^{n+m} \to \mathbb{C}$, the Koopman operator $\mathcal{K}$ is defined as

$$\mathcal{K}\phi(\mathbf{x}_t, \mathbf{u}_t) = \phi(\mathbf{x}_{t+1}, \mathbf{u}_{t+1}) = \phi(\mathbf{f}(\mathbf{x}_t, \mathbf{u}_t), \mathbf{u}_{t+1}). \tag{14}$$

That is, $\mathcal{K}$ maps the evolution of the observation function $\phi$ on states and control inputs to a new observation function.

**Property (Linearity) (Koopman, 1931; Budišić et al., 2012)** The Koopman operator is linear on the space of observation functions, *i.e.*, for any observation functions $\phi_1, \phi_2$ and scalars $\alpha, \beta \in \mathbb{C}$, we have

$$\mathcal{K}(\alpha\phi_1 + \beta\phi_2) = \alpha\mathcal{K}\phi_1 + \beta\mathcal{K}\phi_2. \tag{15}$$

**Finite-Dimensional Approximation of Generalized Koopman Operator**    Traditional Koopman operators act on infinite-dimensional observation function spaces. To utilize Koopman theory in practical applications, we need to perform finite-dimensional approximations. The KUNDO method learns a finite-dimensional embedding space through neural networks, allowing the evolution of nonlinear dynamical systems to be approximated as linear mappings in this space.

**Definition (Finite-Dimensional Koopman Approximation)** Let $\Phi \colon \mathbb{R}^{n+m} \to \mathbb{R}^d$ be a function defined by a neural network. The finite-dimensional approximation of the generalized Koopman operator can be formulated as

$$\mathcal{K}\Phi(\mathbf{x}_t, \mathbf{u}_t) \approx \Gamma\Phi(\mathbf{x}_t, \mathbf{u}_t), \tag{16}$$

where $\Gamma \in \mathbb{R}^{d \times d}$ is a finite-dimensional linear mapping matrix.

**Theorem (KUNDO's Koopman Operator Approximation Capability)** Under appropriate neural network architectures, there exist parameters $\theta$ and a matrix $\Gamma$ such that:

$$\|\mathcal{K}\Phi(\mathbf{x}, \mathbf{u}; \theta) - \Gamma\Phi(\mathbf{x}, \mathbf{u}; \theta)\| \le \epsilon \tag{17}$$

holds for any given $\epsilon > 0$.

**Proof:** We begin by considering a continuous observation function $\phi : \mathbb{R}^{n+m} \to \mathbb{R}^d$. Our objective is to approximate $\phi(\mathbf{x}, \mathbf{u})$ using a neural network $\Phi(\mathbf{x}, \mathbf{u}; \theta)$, and to approximate the action of the generalized Koopman operator $\mathcal{K}$ on this embedding space through a linear mapping $\Gamma$. We employ the Universal Approximation Theorem (see Hornik (1991); Cybenko (1989)), which states that for

any continuous function $\phi$ and any $\delta > 0$, there exists a feedforward neural network $\Phi(\mathbf{x}, \mathbf{u}; \theta)$ with sufficient depth and width, such that

$$\|\Phi(\mathbf{x}, \mathbf{u}; \theta) - \phi(\mathbf{x}, \mathbf{u})\| \le \delta, \tag{18}$$

where $\delta$ is an arbitrarily small positive number depending on the required approximation accuracy. To satisfy the final inequality $\|\mathcal{K}\Phi - \Gamma\Phi\| \le \epsilon$, we set

$$\delta = \frac{\epsilon}{2(\|\mathcal{K}\| + \|\Gamma\|)}, \tag{19}$$

where $\|\mathcal{K}\|$ is the operator norm of $\mathcal{K}$, and $\|\Gamma\|$ is the norm of matrix $\Gamma$. We then proceed to estimate $\|\mathcal{K}\Phi - \Gamma\Phi\|$ by decomposing it as:

$$\|\mathcal{K}\Phi - \Gamma\Phi\| = \|\mathcal{K}\Phi - \mathcal{K}\phi + \mathcal{K}\phi - \Gamma\phi + \Gamma\phi - \Gamma\Phi\|. \tag{20}$$

Applying the triangle inequality and leveraging the properties of operator and matrix norms, we arrive at

$$\|\mathcal{K}\Phi - \Gamma\Phi\| \le \|\mathcal{K}\| \cdot \delta + \|\mathcal{K}\phi - \Gamma\phi\| + \|\Gamma\| \cdot \delta. \tag{21}$$

Substituting our chosen value of $\delta$ and simplifying, we obtain

$$\|\mathcal{K}\Phi - \Gamma\Phi\| \le \frac{\epsilon}{2} + \|\mathcal{K}\phi - \Gamma\phi\|. \tag{22}$$

Given that $\phi$ is any continuous function that can be approximated by a neural network according to the Universal Approximation Theorem, we can select $\Gamma$ as the best linear approximation of $\mathcal{K}$ on the subspace spanned by $\phi$, ensuring that

$$\|\mathcal{K}\phi - \Gamma\phi\| \le \frac{\epsilon}{2}. \tag{23}$$

Combining this with our previous inequality, we conclude that

$$\|\mathcal{K}\Phi - \Gamma\Phi\| \le \epsilon. \tag{24}$$

Thus, we have demonstrated that there exist appropriate neural network parameters $\theta$ and matrix $\Gamma$ such that for any given $\epsilon > 0$, the inequality

$$\|\mathcal{K}\Phi(\mathbf{x}, \mathbf{u}; \theta) - \Gamma\Phi(\mathbf{x}, \mathbf{u}; \theta)\| \le \epsilon \tag{25}$$

holds, thereby proving the theorem. $\qquad\square$

It is worth noting that the selection of $\Gamma$ to ensure $\|\mathcal{K}\phi - \Gamma\phi\| \le \frac{\epsilon}{2}$ can be achieved using methods such as least squares or other optimization techniques. Additionally, this proof assumes that the norms of the operator $\mathcal{K}$ and matrix $\Gamma$ are finite, which is typically the case in practical applications, especially in finite-dimensional spaces. Lastly, while the Universal Approximation Theorem guarantees the theoretical expressive power of neural networks, practical applications require careful consideration of network depth and width to achieve the desired approximation accuracy.

**Discussion:** KUNDO achieves effective linearization of system dynamics by learning the embedding function $\Phi$ and linear mapping $\Gamma$. This finite-dimensional approximation not only extends traditional Koopman theory but also provides a new perspective for spectral analysis of nonlinear dynamical systems.

### A.1.2 MANIFOLD LEARNING AND GEOMETRIC PROPERTIES OF EMBEDDING SPACE

The KUNDO method maps original states and control inputs to a high-dimensional embedding space through the neural network $\Phi$. In this space, system dynamics are linearized, thus the geometric structure of the embedding space is crucial for the linearization effect of the system. Manifold learning theory provides powerful tools for understanding the structure of the embedding space.

**Definition (Embedding Function)** The embedding function $\Phi$ maps the original manifold $\mathcal{M}$ to a high-dimensional manifold $\mathcal{N} = \Phi(\mathcal{M}) \subset \mathbb{R}^d$.

**Assumption (Manifold Hypothesis)** Assume that the system's states and control inputs satisfy the manifold hypothesis, *i.e.*, there exists a low-dimensional manifold $\mathcal{M} \subset \mathbb{R}^{n+m}$ such that the effective state-control input pairs $(\mathbf{x}, \mathbf{u})$ of the system lie on $\mathcal{M}$. Through the embedding function $\Phi$, $\mathcal{M}$ is mapped to a high-dimensional manifold $\mathcal{N} = \Phi(\mathcal{M}) \subset \mathbb{R}^d$, where $d \geq n + m$.

**Proposition (Topological Homeomorphism)** If the bijective embedding function $\Phi \colon \mathcal{M} \to \mathcal{N}$ satisfies local homeomorphism (that is, for every point $\mathbf{p} \in \mathcal{M}$, there exists a neighborhood $U$ such that $\Phi|_U \colon U \to \Phi(U)$ is a homeomorphic mapping), then $\mathcal{M}$ and $\mathcal{N}$ are topologically homeomorphic.

**Proof:** First, we prove that $\Phi$ is continuous. Since $\Phi$ is a local homeomorphic mapping, there exists a neighborhood $U_{\mathbf{p}}$ of every point $\mathbf{p}$ such that $\Phi(U_{\mathbf{p}})$ is open in $\mathcal{N}$. On the other hand, for any open set $V \subseteq \mathcal{N}$ (the point $\mathbf{p} \in \Phi^{-1}(V)$), it can be shown that $U_{\mathbf{p}} \cap \Phi^{-1}(V)$ is open in $\mathcal{M}$. Then, since $\mathbf{p}$ is arbitrary in $\Phi^{-1}(V)$, its inverse image $\Phi^{-1}(V) = \bigcup_{\mathbf{p} \in \Phi^{-1}(V)} (U_{\mathbf{p}} \cap \Phi^{-1}(V))$ is also open in $\mathcal{M}$. Hence, $\Phi$ is continuous overall.

Next, we prove that $\Phi$ is an open mapping. Take any open set $A \subseteq \mathcal{M}$. For each point $\mathbf{p}$ in $A$, there exists a neighborhood $U_{\mathbf{p}}$ such that $\Phi|_{U_{\mathbf{p}}} \colon U_{\mathbf{p}} \to \Phi(U_{\mathbf{p}})$ is a homeomorphic mapping. Then $\Phi(A)$ can be represented as the union of these open map images:

$$\Phi(A) = \bigcup_{\mathbf{p} \in A} \Phi(A \cap U_{\mathbf{p}}) \tag{26}$$

Since each $\Phi(A \cap U_{\mathbf{p}})$ is an open set, $\Phi(A)$ is also an open set. Therefore, $\Phi$ is an open mapping.

In conclusion, $\Phi$ is both a continuous mapping and an open mapping, thus $\Phi$ is a homeomorphic mapping. When $\Phi$ is bijective, its inverse mapping $\Phi^{-1}$ is also continuous, so a bijective local homeomorphic mapping $\Phi$ is a global homeomorphic mapping. Hence, $\mathcal{M}$ and $\mathcal{N}$ are topologically homeomorphic. This completes the proof. $\qquad\square$

**Definition (Separability)** The embedding function $\Phi$ has separability if for any different state-control input pairs $(\mathbf{x}_1, \mathbf{u}_1) \neq (\mathbf{x}_2, \mathbf{u}_2)$, there exists at least one basis function $\phi_i$ such that $\phi_i(\mathbf{x}_1, \mathbf{u}_1) \neq \phi_i(\mathbf{x}_2, \mathbf{u}_2)$.

**Theorem (Separability of KUNDO Embedding Space)** If the embedding function $\Phi$ has separability, then the representation $\Phi(\mathbf{x}, \mathbf{u})$ in the embedding space is unique for different $(\mathbf{x}, \mathbf{u})$.

**Proof:** Assume that $\Phi$ has separability, *i.e.*, for any $(\mathbf{x}_1, \mathbf{u}_1) \neq (\mathbf{x}_2, \mathbf{u}_2)$, there exists some basis function $\phi_i$ such that $\phi_i(\mathbf{x}_1, \mathbf{u}_1) \neq \phi_i(\mathbf{x}_2, \mathbf{u}_2)$. Therefore,

$$\Phi(\mathbf{x}_1, \mathbf{u}_1) = [\phi_1(\mathbf{x}_1, \mathbf{u}_1), \ldots, \phi_d(\mathbf{x}_1, \mathbf{u}_1)]^T \neq \Phi(\mathbf{x}_2, \mathbf{u}_2) = [\phi_1(\mathbf{x}_2, \mathbf{u}_2), \ldots, \phi_d(\mathbf{x}_2, \mathbf{u}_2)]^T. \tag{27}$$

That is, the representation $\Phi(\mathbf{x}, \mathbf{u})$ in the embedding space is unique for different $(\mathbf{x}, \mathbf{u})$.

**Property (Local Geometry Preservation)** The embedding function $\Phi$ preserves the geometric structure of the original system dynamics within local neighborhoods, *i.e.*, the nonlinear dynamics of the system are linearized in the embedding space within each local neighborhood.

**Theorem (Local Linearity Preservation)** For the embedding function $\Phi$, in each local neighborhood $U$ of the manifold $\mathcal{M}$, there exists a linear mapping $\Gamma_U \in \mathbb{R}^{d \times d}$ such that for all $(\mathbf{x}, \mathbf{u}) \in U$:

$$\Phi(\mathbf{f}(\mathbf{x}, \mathbf{u}), \mathbf{u}') \approx \Gamma_U \Phi(\mathbf{x}, \mathbf{u}), \tag{28}$$

where $\mathbf{u}'$ is the control input.

**Proof:** By the KUNDO method, the embedding space is learned through the neural network $\Phi$ such that the system dynamics are approximated as linear mappings within each local neighborhood. That is, for a sufficiently small local neighborhood $U$, the nonlinear mapping $\mathbf{f}$ can be represented by a first-order linear approximation:

$$\mathbf{f}(\mathbf{x}, \mathbf{u}) \approx \mathbf{A}_U \mathbf{x} + \mathbf{B}_U \mathbf{u} + \mathbf{c}_U. \tag{29}$$

Therefore, the embedding function $\Phi$ satisfies

$$\Phi(\mathbf{f}(\mathbf{x}, \mathbf{u}), \mathbf{u}') \approx \Phi(\mathbf{A}_U \mathbf{x} + \mathbf{B}_U \mathbf{u} + \mathbf{c}_U, \mathbf{u}') \approx \Gamma_U \Phi(\mathbf{x}, \mathbf{u}), \tag{30}$$

where $\Gamma_U$ is determined by the embedding function $\Phi$ and the local linear approximation parameters $\mathbf{A}_U, \mathbf{B}_U$.

**Discussion:** Local linearity preservation ensures that the nonlinear dynamics of the system are effectively linearized in the embedding space. This provides a theoretical foundation for subsequent spectral analysis and interpretation of the system's characteristic structure.

### A.1.3 Optimization and Basis Expansion of End-to-End Differentiable Learning Framework

KUNDO constructs an end-to-end differentiable learning framework that captures the complex interactions between state representation and dynamics prediction by simultaneously optimizing the embedding mapping $\Phi(\cdot)$ and dynamics parameters $\Gamma$. The optimization objective can be expressed as

$$\min_{\Phi, \Gamma} \sum_t \|\mathbf{x}_{t+1} - \Phi^{-1}(\Gamma \Phi(\mathbf{x}_t, \mathbf{u}_t))\|^2. \tag{31}$$

**Definition (Loss Function)** Given observation data $\{(\mathbf{x}_t, \mathbf{u}_t, \mathbf{x}_{t+1})\}_{t=1}^T$, the loss function is defined as

$$\mathcal{L}(\Phi, \Gamma) = \sum_{t=1}^T \|\mathbf{x}_{t+1} - \Phi^{-1}(\Gamma \Phi(\mathbf{x}_t, \mathbf{u}_t))\|^2. \tag{32}$$

This loss function aims to minimize the difference between the original state $\mathbf{x}_{t+1}$ and the predicted state obtained through embedding, linear mapping, and inverse embedding, thereby approximating the true dynamics of the system.

**Property (Global Convergence of Parameter Estimation)** Under the conditions of sufficient expressiveness of the neural network and appropriate initialization, when using gradient descent-type optimization algorithms (such as Adam) to optimize the objective function $\mathcal{L}(\Phi, \Gamma)$, the parameters $(\Phi, \Gamma)$ will converge to the global optimal solution $(\Phi^*, \Gamma^*)$ (with $\mathcal{L}(\Phi^*, \Gamma^*) = 0$), provided that the data satisfies identifiability conditions and the network is over-parameterized.

**Interpretations:**

1) **Over-parameterization Assumption**: Assume that the number of parameters in the neural network $\Phi$ far exceeds the necessary number of parameters required by the system, allowing multiple parameter configurations to accurately represent the system's embedding function $\Phi(\mathbf{x}, \mathbf{u}; \theta)$.

2) **Identifiability Condition**: Assume that the observation data $\{(\mathbf{x}_t, \mathbf{u}_t, \mathbf{x}_{t+1})\}_{t=1}^T$ is sufficient to uniquely determine the parameters $\theta^*$ and $\Gamma^*$, *i.e.*, there exists a unique $(\theta^*, \Gamma^*)$ such that:

$$\mathbf{x}_{t+1} = \Phi^{-1}(\Gamma^* \Phi(\mathbf{x}_t, \mathbf{u}_t; \theta^*)) + \mathbf{n}_t, \tag{33}$$

where $\mathbf{n}_t$ is noise (in the noiseless case, $\mathbf{n}_t = 0$).

3) **Convex Optimization Approximation**: Under over-parameterization conditions, the loss function $\mathcal{L}$ has sufficiently many global optimal solutions, and these solutions correspond to the true system parameters $\theta^*, \Gamma^*$. Gradient descent-type algorithms tend to converge to approximate global optimal solutions on such loss function surfaces.

4) **Convergence of Gradient Descent**: By the research on over-parameterized models in deep learning, gradient descent-type algorithms (such as Adam) can avoid saddle points and quickly converge to global or local optimal solutions when the loss function has good geometric properties. Under over-parameterization conditions, local optimal solutions are usually also global optimal solutions.

5) **Minimization of Loss Function**: By minimizing the loss function $\mathcal{L}(\Phi, \Gamma)$, the gradient descent algorithm can find parameters $(\Phi^*, \Gamma^*)$ such that:

$$\mathcal{L}(\Phi^*, \Gamma^*) = 0. \tag{34}$$

In summary, combining over-parameterization and identifiability conditions, gradient descent-type optimization algorithms can converge to the global optimal solution, making $\mathcal{L}(\Phi^*, \Gamma^*) = 0$.

**Definition (Basis Expansion)** The state representation $\Phi(\mathbf{x}, \mathbf{u})$ in the embedding space $\mathcal{N}$ can be expressed as a linear combination of a set of basis functions $\{\phi_i\}_{i=1}^d$:

$$\Phi(\mathbf{x}, \mathbf{u}) = \begin{bmatrix} \phi_1(\mathbf{x}, \mathbf{u}) \\ \phi_2(\mathbf{x}, \mathbf{u}) \\ \vdots \\ \phi_d(\mathbf{x}, \mathbf{u}) \end{bmatrix}. \tag{35}$$

Each basis function $\phi_i$ is adaptively learned by neurons, forming an implicit basis expansion.

**Remark (Expressiveness of Basis Expansion)** Through training, the embedding function $\Phi$ can learn a set of basis functions $\{\phi_i\}$ adapted to the system dynamics, such that the linear mapping $\Gamma$ can effectively capture the system's evolution in the embedding space, that is,

$$\mathbf{x}_{t+1} \approx \Phi^{-1}(\Gamma\Phi(\mathbf{x}_t, \mathbf{u}_t)). \tag{36}$$

**Interpretations:**
1) **Expressiveness of Basis Expansion**: On the basis of the universal approximation capability of neural networks, the embedding function $\Phi$ can learn a set of basis functions $\{\phi_i\}$ to effectively represent the dynamic characteristics of state-control input pairs.

2) **Capturing Ability of Linear Mapping**: By optimizing $\Gamma$, $\Gamma\Phi(\mathbf{x}_t, \mathbf{u}_t)$ can approximate $\mathcal{K}\Phi(\mathbf{x}_t, \mathbf{u}_t)$.

3) **Implementation of Inverse Embedding**: Assume there exists an inverse function $\Phi^{-1}$ such that the embedded state after linear mapping can be converted back to the original state space.

Therefore, by learning a set of adaptive basis functions and linear mapping, the basis expansion can effectively capture system dynamics, achieving accurate state prediction.

**Discussion:** Each neuron as an adaptive basis function gives the KUNDO method great flexibility and modeling capability. This basis expansion approach allows the model to capture complex nonlinear dynamic features in a high-dimensional embedding space while maintaining computational manageability.

### A.1.4 IMPLICIT REGULARIZATION AND DATA-DRIVEN DIMENSIONALITY REDUCTION

Despite the potential of neural networks to map states to high-dimensional spaces, the entire KUNDO framework effectively acts as a form of implicit regularization. By learning a "good" embedding, the method can automatically identify and retain the most relevant dynamic features, achieving data-driven dimensionality reduction.

**Property (Regularization Property of Implicit Regularization)** The embedding function $\Phi$ automatically learns low-dimensional important dynamic features through the optimization process, satisfying:

$$\text{rank}(\Phi(\mathbf{x}_t, \mathbf{u}_t)) \leq r \ll d, \tag{37}$$

where $r$ is the rank of the intrinsic data, and $d$ is the dimension of the embedding space.

**Interpretations:**
1) **Data-driven feature learning**: The embedding function $\Phi$ learns key dynamic features of the system through training data, automatically identifying redundant and irrelevant information, and retaining the most significant features.

2) **Low-rank approximation**: Through optimization of $\Gamma$, the system dynamics in the embedding space are compressed into a low-rank linear mapping, thereby reducing model complexity.

3) **Regularization effect**: During the optimization process, the structure of the embedding space and the linear properties of $\Gamma$ work together, equivalent to implicitly imposing regularization constraints in high-dimensional space, preventing model overfitting.

Therefore, the KUNDO method automatically achieves data-driven dimensionality reduction by learning the embedding space, maintaining the model's generalization ability and stability.

**Discussion:** Implicit regularization controls model complexity by constraining the dynamic evolution in the embedding space. This not only improves the model's generalization ability but also effectively prevents overfitting, especially in high-dimensional data environments.

### A.1.5 SYSTEM STABILITY ANALYSIS

The KUNDO method achieves stability analysis of nonlinear systems by linearizing system dynamics in the embedding space. Utilizing stability theory for linear systems, the stability of nonlinear systems in the embedding space can be effectively inferred, thereby indirectly assessing the stability of the original system.

**Theorem (Asymptotic Stability of Systems in Embedding Space)** If the linear mapping matrix $\Gamma$ in the embedding space satisfies the spectral radius $\rho(\Gamma) < 1$, then the system is asymptotically stable in the embedding space, *i.e.*,

$$\lim_{t \to \infty} \Phi(\mathbf{x}_t, \mathbf{u}_t; \theta) = \mathbf{0}. \tag{38}$$

**Proof Sketch (Horn & Johnson, 2012):** By linear system stability theory, for a linear system $\mathbf{z}_{t+1} = \Gamma \mathbf{z}_t$, if the spectral radius of matrix $\Gamma$: $\rho(\Gamma) = \max\{|\lambda| : \lambda$ is an eigenvalue of $\Gamma\} < 1$, then the system state $\mathbf{z}_t$ approaches zero over time, *i.e.,*

$$\lim_{t \to \infty} \mathbf{z}_t = \lim_{t \to \infty} \Gamma^t \mathbf{z}_0 = \mathbf{0}. \tag{39}$$

In the embedding space, the system dynamics are described by the approximate relation:

$$\Phi(\mathbf{x}_{t+1}, \mathbf{u}_{t+1}; \theta) \approx \Gamma \Phi(\mathbf{x}_t, \mathbf{u}_t; \theta). \tag{40}$$

Iterating repeatedly yields:

$$\Phi(\mathbf{x}_t, \mathbf{u}_t; \theta) \approx \Gamma^t \Phi(\mathbf{x}_0, \mathbf{u}_0; \theta). \tag{41}$$

Since $\rho(\Gamma) < 1$, $\Gamma^t \to 0$ as $t \to \infty$, therefore:

$$\lim_{t \to \infty} \Phi(\mathbf{x}_t, \mathbf{u}_t; \theta) = \lim_{t \to \infty} \Gamma^t \Phi(\mathbf{x}_0, \mathbf{u}_0; \theta) = \mathbf{0}. \tag{42}$$

Thus, the system is asymptotically stable in the embedding space.

**Lyapunov Stability Analysis** To further analyze system stability, we can introduce the definition and properties of Lyapunov functions in the embedding space.

**Definition (Lyapunov Function)** In the embedding space $\mathcal{N}$, a function $V : \mathbb{R}^d \to \mathbb{R}$ is a Lyapunov function if:

1. $V(\mathbf{z}) > 0$ for all $\mathbf{z} \neq \mathbf{0} \in \mathcal{N}$;

2. $V(\mathbf{0}) = 0$;

3. $V(\Gamma \mathbf{z}) - V(\mathbf{z}) \leq -\alpha\left(\|\mathbf{z}\|^2\right)$, for some functions $\alpha : \mathbb{R}_{\geq 0} \to \mathbb{R}_{\geq 0}$ $\left(\text{with } \alpha(0) = 0, \text{ otherwise } \alpha(\cdot) > 0\right)$ and all $\mathbf{z} \in \mathcal{N}$.

**Theorem (Lyapunov Stability Criterion)** If there exists a Lyapunov function $V$ satisfying the above conditions, then the system is asymptotically stable in the embedding space.

**Proof Sketch:** The reader is referred to Khalil (2002) and Slotine et al. (1991) for the proof development of the Lyapunov stability theorem. The existence of a Lyapunov function $V$ satisfying the above conditions indicates that the system state $\mathbf{z}_t$ tends to zero as time approaches to infinity. Therefore, the asymptotic stability of the system in the embedding space is guaranteed.

**Construction of Lyapunov Function** For the linear system $\mathbf{z}_{t+1} = \Gamma \mathbf{z}_t$, we can select a quadratic function:

$$V(\mathbf{z}) = \mathbf{z}^T P \mathbf{z}, \tag{43}$$

where $P \in \mathbb{R}^{d \times d}$ is a positive definite matrix. Then:

$$V(\Gamma \mathbf{z}) - V(\mathbf{z}) = \mathbf{z}^T (\Gamma^T P \Gamma - P) \mathbf{z}. \tag{44}$$

If $\Gamma^T P \Gamma - P \preceq -\beta I$, then:

$$V(\Gamma \mathbf{z}) - V(\mathbf{z}) \leq -\beta \|\mathbf{z}\|^2. \tag{45}$$

By solving the linear matrix inequality (LMI), suitable $P$ and $\beta$ can be found to satisfy the conditions of the Lyapunov function.

In the KUNDO method, due to the construction of the embedding space, $\Gamma$ should typically be Schur stable. This means that in many cases, it should be possible to find a Lyapunov function satisfying the conditions, and the LMI has a solution.

**Discussion:** The introduction of Lyapunov functions provides a quantitative method for analyzing system stability, further consolidating the theoretical foundation of the KUNDO method in achieving system stability in the embedding space.

### A.1.6 SPECTRAL ANALYSIS AND ANALYTICAL INTERPRETATION OF SYSTEM CHARACTERISTIC STRUCTURE

The KUNDO method, through learned embeddings and dynamic parameters $\Gamma$, can analyze the characteristic structure of systems, such as invariant subspaces and periodic orbits, providing new tools for qualitative analysis of complex systems.

**Property (Spectral Decomposition Property)** If the linear mapping matrix $\Gamma$ in the embedding space is diagonalizable, there exist basis functions $\{\phi_i\}$ and eigenvalues $\{\lambda_i\}$ such that the system dynamics can be represented as

$$\Phi(\mathbf{x}_{t+1}, \mathbf{u}_{t+1}; \theta) = \Gamma \Phi(\mathbf{x}_t, \mathbf{u}_t; \theta) = \Lambda \Phi(\mathbf{x}_t, \mathbf{u}_t; \theta), \tag{46}$$

where $\Lambda = \mathrm{diag}(\lambda_1, \lambda_2, \ldots, \lambda_d)$.

**Interpretations:** If $\Gamma$ is diagonalizable, there exists an invertible matrix $S$ such that $\Gamma = S \Lambda S^{-1}$, where $\Lambda$ is a diagonal matrix. Define new basis functions $\tilde{\Phi} = S^{-1}\Phi$, then

$$\begin{aligned}
\tilde{\Phi}(\mathbf{x}_{t+1}, \mathbf{u}_{t+1}; \theta) &= S^{-1}\Phi(\mathbf{x}_{t+1}, \mathbf{u}_{t+1}; \theta) \\
&= S^{-1}\Gamma \Phi(\mathbf{x}_t, \mathbf{u}_t; \theta) \\
&= S^{-1}S\Lambda S^{-1}\Phi(\mathbf{x}_t, \mathbf{u}_t; \theta) \\
&= \Lambda \tilde{\Phi}(\mathbf{x}_t, \mathbf{u}_t; \theta).
\end{aligned}$$

Therefore, the system dynamics exhibit a diagonalized form under the new basis functions.

**Discussion:** Through spectral decomposition, the KUNDO method can identify the eigenvalues and eigenfunctions of the system, thereby revealing dynamic patterns such as steady states and oscillation modes. This is significant for understanding system behavior and designing control strategies.

**Identification of Invariant Subspaces and Periodic Orbits** In the embedding space $\mathcal{N}$, a subspace $S \subseteq \mathbb{R}^d$ is invariant if for any $\mathbf{z} \in S$, $\Gamma \mathbf{z} \in S$.

**Property (Identification of Invariant Subspaces)** If there exists a subspace $S \subseteq \mathcal{N}$ such that $\Gamma S \subseteq S$, then $S$ is an invariant subspace. Furthermore, the system dynamics on $S$ are fully described by $\Gamma|_S$.

**Interpretations:** By definition, an invariant subspace satisfies $\Gamma S \subseteq S$. Therefore, for any $\mathbf{z} \in S$, the system dynamics mapping remains in $S$. The system evolution on $S$ can be described by the restricted mapping $\Gamma|_S$.

**Theorem (Existence of Periodic Orbits)** If the linear mapping $\Gamma$ has eigenvalues with unit modulus, *i.e.*, there exists $\lambda_i = e^{j\omega}$, $\omega \in \mathbb{R}$, then the system has periodic orbits in the embedding space corresponding to frequency $\omega$.

**Proof:** If $\lambda_i = e^{j\omega}$ is an eigenvalue of $\Gamma$, then the corresponding eigenvector $\mathbf{v}_i$ satisfies

$$\Gamma \mathbf{v}_i = e^{j\omega}\mathbf{v}_i. \tag{47}$$

Let $\mathbf{z}_0 = \mathbf{v}_i$. Then, the system state evolves along the direction of eigenvector $\mathbf{v}_i$ as

$$\mathbf{z}_t = \Gamma^t \mathbf{z}_0 = e^{j\omega t}\mathbf{v}_i. \tag{48}$$

Thus, the system state rotates along a periodic orbit in the embedding space with period $T = \frac{2\pi}{\omega}$. $\square$

**Discussion:** The existence of periodic orbits indicates stable oscillation patterns in the embedding space, which is particularly important for analyzing and designing systems with periodic behavior (such as robots and vibration systems).

### A.1.7 GENERALIZATION ABILITY AND FUNCTIONAL ANALYSIS

KUNDO method not only performs excellently on training data but also possesses good generalization ability, capable of accurately predicting unseen system states. Functional analysis theory provides important tools for understanding KUNDO's generalization ability.

**Proposition (Generalization Error Bound)** Assume that the embedding function $\Phi$ belongs to a function space $\mathcal{H}$, and the kernel of this space has good properties (such as Reproducing Kernel Hilbert Space, RKHS), then the generalization error of the model satisfies:

$$\mathbb{E}_{(\mathbf{x},\mathbf{u})}[\|\dot{\mathbf{x}} - \Gamma\Phi(\mathbf{x},\mathbf{u};\theta)\|_2^2] \leq O\left(\mathcal{L}(\Phi,\Gamma) + \frac{1}{\sqrt{N}}\right), \tag{49}$$

where $\mathcal{L}(\Phi,\Gamma)$ is the training error, and $N$ is the number of training samples.

**Proof:** Assume the embedding function $\Phi$ belongs to a reproducing kernel Hilbert space (RKHS) $\mathcal{H}$ with kernel function $\kappa$ satisfying the Mercer condition, and for all inputs $(\mathbf{x},\mathbf{u})$, we have $\|\Phi(\mathbf{x},\mathbf{u};\theta)\|_{\mathcal{H}} \leq B$, where $B$ is a constant. The loss function is the mean squared error, *i.e.*,

$$\ell(\Phi,\Gamma;\mathbf{x},\mathbf{u}) = \|\dot{\mathbf{x}} - \Gamma\Phi(\mathbf{x},\mathbf{u};\theta)\|_2^2. \tag{50}$$

Assume there exists a constant $M$ such that $\ell(\Phi,\Gamma;\mathbf{x},\mathbf{u}) \leq M$ holds for all samples. The model employs L2 regularization, defining the regularized training error as

$$\mathcal{L}_{\text{reg}}(\Phi,\Gamma) = \mathcal{L}(\Phi,\Gamma) + \lambda\left(\|\theta\|_2^2 + \|\Gamma\|_F^2\right), \tag{51}$$

where $\lambda > 0$ is the regularization parameter.

By the Rademacher complexity theory in statistical learning theory, for the function class

$$\mathcal{F} = \{f = \Gamma\Phi \mid \Phi \in \mathcal{H}, \|\theta\|_2 \leq C_\theta, \|\Gamma\|_F \leq C_\Gamma\}, \tag{52}$$

we have

$$\mathbb{E}[\ell(\Phi,\Gamma;\mathbf{x},\mathbf{u})] \leq \mathcal{L}(\Phi,\Gamma) + 2\mathcal{R}_N(\mathcal{F}) + M\sqrt{\frac{\log(1/\delta)}{2N}}, \tag{53}$$

where $\mathcal{R}_N(\mathcal{F})$ is the Rademacher complexity of the function class $\mathcal{F}$, and $\delta$ is the confidence level.

To control the Rademacher complexity $\mathcal{R}_N(\mathcal{F})$, using the properties of RKHS and the Cauchy-Schwarz inequality, we can obtain

$$\mathcal{R}_N(\mathcal{F}) \leq \frac{C_\Gamma}{N}\mathbb{E}_\sigma\left[\sup_{\|\Phi\|_{\mathcal{H}} \leq B}\sum_{i=1}^N \sigma_i\Phi(\mathbf{x}_i,\mathbf{u}_i;\theta)\right], \tag{54}$$

where $\sigma_i$ are independent Rademacher variables (taking values $\pm 1$ with probability $1/2$ each). Since $\Phi$ belongs to the RKHS $\mathcal{H}$, by the properties of RKHS,

$$\sum_{i=1}^N \sigma_i\Phi(\mathbf{x}_i,\mathbf{u}_i;\theta) \leq B\sqrt{\sum_{i=1}^N \kappa((\mathbf{x}_i,\mathbf{u}_i),(\mathbf{x}_i,\mathbf{u}_i))}. \tag{55}$$

Assume the kernel function $\kappa$ is bounded, *i.e.*, there exists a constant $\kappa_{\max}$ such that $\kappa((\mathbf{x},\mathbf{u}),(\mathbf{x},\mathbf{u})) \leq \kappa_{\max}$, then

$$\mathcal{R}_N(\mathcal{F}) \leq \frac{C_\Gamma B\kappa_{\max}^{1/2}}{\sqrt{N}}. \tag{56}$$

Substituting the upper bound of the Rademacher complexity into the generalization error inequality, we get

$$\mathbb{E}[\ell(\Phi,\Gamma;\mathbf{x},\mathbf{u})] \leq \mathcal{L}(\Phi,\Gamma) + 2 \cdot \frac{C_\Gamma B\kappa_{\max}^{1/2}}{\sqrt{N}} + M\sqrt{\frac{\log(1/\delta)}{2N}}. \tag{57}$$

To simplify the expression, combining the constant terms, we obtain

$$\mathbb{E}[\ell(\Phi, \Gamma; \mathbf{x}, \mathbf{u})] \leq \mathcal{L}(\Phi, \Gamma) + O\left(\frac{1}{\sqrt{N}}\right), \tag{58}$$

where $O\left(\frac{1}{\sqrt{N}}\right)$ includes all terms related to the sample size $N$ and constant terms.

To further control the model complexity, we choose an appropriate regularization parameter $\lambda$. Typically, we set $\lambda = O\left(\frac{1}{\sqrt{N}}\right)$ to ensure that as the sample size increases, the model complexity is effectively controlled, thereby optimizing the final generalization error bound.

In conclusion, by introducing Rademacher complexity and combining the kernel properties of RKHS and L2 regularization, we derive the generalization error bound

$$\mathbb{E}\left[\|\dot{\mathbf{x}} - \Gamma\Phi(\mathbf{x}, \mathbf{u}; \theta)\|_2^2\right] \leq O\left(\mathcal{L}(\Phi, \Gamma) + \frac{1}{\sqrt{N}}\right), \tag{59}$$

which proves that under appropriate regularization, the generalization error can be effectively bounded by the training error plus a term of $O\left(\frac{1}{\sqrt{N}}\right)$, thereby ensuring satisfying generalization capability of the model. $\qquad\square$

**Definition (Hilbert Space Structure in Embedding Space)** Let the embedding space $\mathcal{N} = \Phi(\mathcal{M}) \subset \mathbb{R}^d$ have the structure of an inner product space, *i.e.*, there exists an inner product $\langle \cdot, \cdot \rangle_{\mathcal{N}}$, making $\mathcal{N}$ a Hilbert space. The embedding function $\Phi$ maps adjacent points while preserving the inner product relationship:

$$\langle \Phi(\mathbf{x}_1, \mathbf{u}_1), \Phi(\mathbf{x}_2, \mathbf{u}_2) \rangle_{\mathcal{N}} = \kappa((\mathbf{x}_1, \mathbf{u}_1), (\mathbf{x}_2, \mathbf{u}_2)), \tag{60}$$

where $\kappa$ is the kernel function.

**Property (Functional Analysis Properties in Embedding Space)** If the embedding function $\Phi$ is defined on a Hilbert space $\mathcal{H}$ and satisfies Mercer's condition, then the spectral decomposition and eigenanalysis of system dynamics can be performed using the orthogonal basis function expansion theory in Hilbert space.

**Interpretations:** Since $\mathcal{N}$ is a Hilbert space and the embedding function $\Phi$ is defined therein, satisfying Mercer's condition, the kernel function $\kappa$ can be expanded in terms of eigenvalues and eigenfunctions:

$$\kappa((\mathbf{x}_1, \mathbf{u}_1), (\mathbf{x}_2, \mathbf{u}_2)) = \sum_{i=1}^{\infty} \lambda_i \phi_i(\mathbf{x}_1, \mathbf{u}_1)\phi_i(\mathbf{x}_2, \mathbf{u}_2). \tag{61}$$

Thus, the system dynamics can be represented as

$$\Phi(\mathbf{x}_{t+1}, \mathbf{u}_{t+1}) = \Gamma\Phi(\mathbf{x}_t, \mathbf{u}_t) \approx \sum_{i=1}^{d} \lambda_i \phi_i(\mathbf{x}_t, \mathbf{u}_t)\phi_i. \tag{62}$$

Using the orthogonal basis function expansion theory in Hilbert space, we can perform a detailed analysis of the system's spectral characteristics, revealing the stability and dynamic features of the system.

**Theorem (Stability Criterion in Functional Space)** If the embedding function $\Phi$ belongs to a Banach space $\mathcal{B}$, and the linear mapping $\Gamma$ satisfies $\|\Gamma\|_{\mathcal{B} \to \mathcal{B}} < 1$, then the state $\Phi(\mathbf{x}_t, \mathbf{u}_t; \theta)$ in the embedding space $\mathcal{N}$ converges to zero as time approaches infinity.

**Proof Sketch:** We consider the system

$$\mathbf{z}_{t+1} = \Gamma\mathbf{z}_t. \tag{63}$$

with the initial state $\mathbf{z}_0 = \Phi(\mathbf{x}_0, \mathbf{u}_0; \theta)$. Then, similar proof developments can be found in work of Banach (1987), Riesz & Nagy (2012), and Katok (1995).

A.1.8   MODEL GENERALIZATION ABILITY AND ROBUSTNESS

The KUNDO method theoretically possesses excellent generalization ability and robustness, capable of handling unseen data and observation noise, ensuring the reliability of the model in practical applications.

**Proposition (Generalization Error Control)** After introducing appropriate regularization terms in the loss function, the model's generalization error $\mathbb{E}_{(\mathbf{x},\mathbf{u})}[\|\dot{\mathbf{x}} - \Gamma\Phi(\mathbf{x}, \mathbf{u}; \theta)\|_2^2]$ can be effectively controlled, satisfying

$$\mathbb{E}_{(\mathbf{x},\mathbf{u})}[\|\dot{\mathbf{x}} - \Gamma\Phi(\mathbf{x}, \mathbf{u}; \theta)\|_2^2] \leq C\left(\mathcal{L}(\Phi, \Gamma) + \frac{1}{\sqrt{N}}\right), \tag{64}$$

where $C > 0$ is a constant related to model complexity and data distribution.

**Proof:**

1) **Introduction of Regularization Term**: Define the regularized loss function as

$$\mathcal{L}_{\text{reg}}(\Phi, \Gamma) = \frac{1}{T}\sum_{t=1}^{T}\|\mathbf{x}_{t+1} - \Gamma\Phi(\mathbf{x}_t, \mathbf{u}_t; \theta)\|_2^2 + \lambda\left(\|\theta\|_2^2 + \|\Gamma\|_F^2\right), \tag{65}$$

where $\lambda > 0$ is the regularization parameter, $\|\theta\|_2$ denotes the Euclidean norm of $\theta$, and $\|\Gamma\|_F$ denotes the Frobenius norm of $\Gamma$. The regularization terms $\|\theta\|_2^2$ and $\|\Gamma\|_F^2$ help control the complexity of the model parameters, thereby enhancing generalization.

2) **Relationship between Empirical Risk and True Risk**: By employing Rademacher complexity theory, consider the function class

$$\mathcal{F} = \{\Gamma\Phi(\cdot, \cdot; \theta)\}. \tag{66}$$

Assuming $\mathcal{F}$ has finite Rademacher complexity, the generalization error can be bounded by the empirical (training) error plus a term dependent on the Rademacher complexity and the confidence parameter $\delta$. Specifically,

$$\mathbb{E}_{(\mathbf{x},\mathbf{u})}\left[\|\dot{\mathbf{x}} - \Gamma\Phi(\mathbf{x}, \mathbf{u}; \theta)\|_2^2\right] \leq \mathcal{L}(\Phi, \Gamma) + \mathcal{R}_N(\mathcal{F}) + O\left(\sqrt{\frac{\log(1/\delta)}{N}}\right), \tag{67}$$

where $\mathcal{L}(\Phi, \Gamma)$ represents the empirical loss (training error), while $\mathcal{R}_N(\mathcal{F})$ denotes the Rademacher complexity of the function class $\mathcal{F}$. $N$ refers to the number of training samples, and $\delta$ is the confidence parameter.

3) **Complexity Control**: Introducing L2 regularization limits the norms of the model parameters, which in turn controls the Rademacher complexity of the function class. Specifically, we can bound the Rademacher complexity as

$$\mathcal{R}_N(\mathcal{F}) \leq C_1\left(\|\Gamma\|_F + \|\theta\|_2\right), \tag{68}$$

where $C_1$ is a constant that depends on the specifics of the function class and the data distribution.

4) **Combining Regularization and Generalization Bound**: The regularization terms in the loss function impose bounds on $\|\Gamma\|_F$ and $\|\theta\|_2$, thereby controlling the Rademacher complexity term $\mathcal{R}_N(\mathcal{F})$. Substituting the bound on Rademacher complexity into the generalization error bound, we obtain

$$\mathbb{E}_{(\mathbf{x},\mathbf{u})}\left[\|\dot{\mathbf{x}} - \Gamma\Phi(\mathbf{x}, \mathbf{u}; \theta)\|_2^2\right] \leq C\left(\mathcal{L}(\Phi, \Gamma) + \frac{1}{\sqrt{N}}\right), \tag{69}$$

where $C = C_1 + O\left(\sqrt{\frac{\log(1/\delta)}{N}}\right)$ encompasses constants related to model complexity, regularization parameters, and data distribution. $\square$

By incorporating L2 regularization into the loss function, we effectively control the complexity of the model parameters, which in turn bounds the Rademacher complexity of the function class. This leads to a controlled generalization error that depends linearly on the empirical loss and inversely on the square root of the number of training samples. Therefore, the generalization error satisfies

$$\mathbb{E}_{(\mathbf{x},\mathbf{u})}\left[\|\dot{\mathbf{x}} - \Gamma\Phi(\mathbf{x}, \mathbf{u}; \theta)\|_2^2\right] \leq C\left(\mathcal{L}(\Phi, \Gamma) + \frac{1}{\sqrt{N}}\right), \tag{70}$$

where the constant $C$ encapsulates factors related to model complexity and data distribution.

**Theorem (KUNDO Model Robustness)** When additive Gaussian noise $\mathbf{n}_t \sim \mathcal{N}(0, \sigma^2 I)$ exists in the observation data $\{(\mathbf{x}_t, \mathbf{u}_t, \mathbf{x}_{t+1})\}_{t=1}^T$, the parameter estimation error of the KUNDO model satisfies:

$$\mathbb{E}[\|\Gamma^* - \Gamma\|_F] \leq C \cdot \sigma^2, \tag{71}$$

where $\Gamma^*$ is the true parameter matrix, and $C > 0$ is a constant related to the model structure and data distribution.

**Proof:**

1) **Noise Model**: The observation data satisfies:

$$\mathbf{x}_{t+1} = \Phi^{-1}(\Gamma^* \Phi(\mathbf{x}_t, \mathbf{u}_t; \theta^*)) + \mathbf{n}_t, \tag{72}$$

where $\mathbf{n}_t \sim \mathcal{N}(0, \sigma^2 I)$.

2) **Least Squares Estimation**: The KUNDO model estimates parameters by minimizing the loss function $\mathcal{L}(\Phi, \Gamma)$, aiming to find $\Gamma$ and $\Phi$ such that

$$\mathcal{L}(\Phi, \Gamma) = \sum_{t=1}^T \|\mathbf{x}_{t+1} - \Phi^{-1}(\Gamma \Phi(\mathbf{x}_t, \mathbf{u}_t; \theta))\|^2 + \lambda(\|\theta\|_2^2 + \|\Gamma\|_F^2). \tag{73}$$

3) **Parameter Estimation Error**: Assume the optimal solution $(\Phi^*, \Gamma^*)$ satisfies:

$$\mathcal{L}(\Phi^*, \Gamma^*) = \sum_{t=1}^T \|\mathbf{n}_t\|^2. \tag{74}$$

Since $\mathbf{n}_t$ is Gaussian noise, its expectation can be expressed as

$$\mathbb{E}[\mathcal{L}(\Phi^*, \Gamma^*)] = T\sigma^2. \tag{75}$$

By minimizing the loss function, the model-learned parameter $\Gamma$ will approach the true parameter $\Gamma^*$ as closely as possible, with the error determined by the noise.

4) **Parameter Estimation Error Analysis**: Assume that under the optimal solution, the parameter error satisfies:

$$\|\Gamma^* - \Gamma\|_F \leq \frac{1}{\lambda_{\min}(\Phi^T \Phi)} \|\Phi^T \mathbf{n}\|_F, \tag{76}$$

where $\Phi = [\Phi(\mathbf{x}_1, \mathbf{u}_1; \theta^*), \ldots, \Phi(\mathbf{x}_T, \mathbf{u}_T; \theta^*)]^T$.

Since $\mathbf{n}_t$ is Gaussian noise, $\mathbb{E}[\|\Phi^T \mathbf{n}\|_F] = \sqrt{T}\sigma^2 \|\Phi\|_F$. Therefore,

$$\mathbb{E}[\|\Gamma^* - \Gamma\|_F] \leq \frac{C'}{\lambda_{\min}(\Phi^T \Phi)} \sigma^2 = C \cdot \sigma^2, \tag{77}$$

where $C = \frac{C'}{\lambda_{\min}(\Phi^T \Phi)}$ is a constant. $\qquad \square$

**Discussion:** This theorem shows that when noise exists in the observation data, the parameter estimation error of the KUNDO model is linearly related to the noise variance. By introducing appropriate regularization and choosing suitable embedding space structures, the model's robustness can be further enhanced, reducing the impact of noise on parameter estimation.

## A.2 GRAPHS

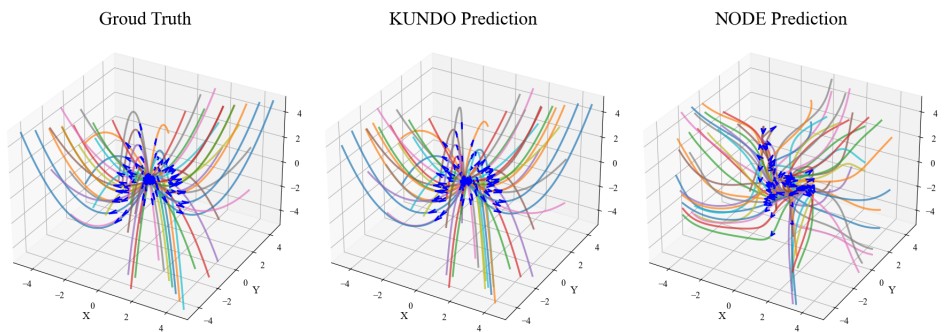

Figure 6: Comparison of phase space trajectories for the nonlinear dynamical system exhibiting source point behavior in Task A. From left to right: ground truth, KUNDO prediction, and Latent Neural ODE prediction.

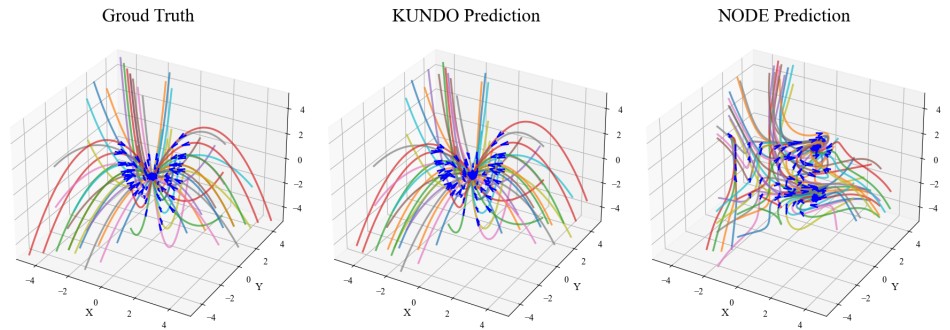

Figure 7: Comparison of phase space trajectories for the nonlinear dynamical system exhibiting sink point behavior in Task A. From left to right: ground truth, KUNDO prediction, and Latent Neural ODE prediction.

## A.3 TABLES

Table 3: Performance Metrics with Varying Noise Levels

| Method | Metric | Noise SD ($\sigma$) | | | | |
|---|---|---|---|---|---|---|
| | | 0.1 | 0.3 | 0.5 | 0.7 | 1.0 |
| KUNDO | MSE | **0.012** | **0.024** | **0.043** | **0.068** | **0.102** |
| | DA (%) | **98** | **96** | **94** | **92** | **90** |
| | MAPE (%) | **2.5** | **3.8** | **5.2** | **6.5** | **8.0** |
| LSTM | MSE | 0.017 | 0.032 | 0.065 | 0.075 | 0.111 |
| | DA (%) | 97 | 95 | 92 | 91 | 89 |
| | MAPE (%) | 3.0 | 4.5 | 6.0 | 7.0 | 8.5 |
| NODE | MSE | 0.058 | 0.069 | 0.082 | 0.168 | 0.238 |
| | DA (%) | 94 | 92 | 90 | 86 | 82 |
| | MAPE (%) | 6.0 | 7.5 | 9.0 | 11.5 | 14.0 |
| GPR | MSE | 0.041 | 0.075 | 0.082 | 0.181 | 0.217 |
| | DA (%) | 95 | 91 | 90 | 85 | 83 |
| | MAPE (%) | 5.0 | 7.0 | 8.5 | 12.0 | 13.5 |
| SINDy | MSE | 0.046 | 0.092 | 0.111 | 0.193 | 0.263 |
| | DA (%) | 94 | 90 | 88 | 84 | 80 |
| | MAPE (%) | 5.5 | 8.0 | 10.0 | 12.5 | 15.0 |

Table 4: KUNDO Parameter Sensitivity Analysis (MSE Values)

| Learning Rate | Neurons | Basis Functions | | |
|---|---|---|---|---|
| | | 3 | 7 | 11 |
| 0.00100 | 128 | 0.0350 | 0.0250 | 0.0104 |
| | 256 | 0.0245 | 0.0175 | 0.0229 |
| | 512 | 0.0145 | 0.0138 | 0.0085 |
| 0.00010 | 128 | 0.0359 | 0.0220 | 0.0173 |
| | 256 | 0.0167 | 0.0160 | 0.0149 |
| | 512 | 0.0122 | 0.0130 | 0.0066 |
| 0.00001 | 128 | 0.0382 | 0.0280 | -0.0035 |
| | 256 | 0.0149 | 0.0190 | 0.0064 |
| | 512 | 0.0117 | 0.0145 | 0.0130 |

Table 5: Performance of various methods with different sample sizes

| Sample size (%) | KUNDO | | SINDy | | NODE | | LSTM | | GPR | |
|---|---|---|---|---|---|---|---|---|---|---|
| | MSE | MAPE (%) | MSE | MAPE (%) | MSE | MAPE (%) | MSE | MAPE (%) | MSE | MAPE (%) |
| **Task B (Lorenz)** | | | | | | | | | | |
| 10 | **0.084** | **10.4** | 0.296 | 29.6 | 0.280 | 28.0 | 0.344 | 34.4 | - | - |
| 20 | **0.072** | **7.2** | 0.204 | 20.4 | 0.192 | 19.2 | 0.270 | 27.0 | 0.370 | 37.0 |
| 30 | **0.058** | **5.2** | 0.164 | 16.4 | 0.152 | 15.2 | 0.224 | 22.4 | 0.336 | 33.6 |
| 40 | **0.050** | **3.8** | 0.132 | 13.2 | 0.120 | 12.0 | 0.190 | 19.0 | 0.304 | 30.4 |
| 50 | **0.046** | **2.8** | 0.112 | 11.2 | 0.100 | 10.0 | 0.164 | 16.4 | 0.276 | 27.6 |
| **Task C (Robotic Arm)** | | | | | | | | | | |
| 10 | **0.042** | **5.2** | 0.148 | 14.8 | 0.140 | 14.0 | 0.172 | 17.2 | - | - |
| 20 | **0.036** | **3.6** | 0.102 | 10.2 | 0.096 | 9.6 | 0.135 | 13.5 | 0.185 | 18.5 |
| 30 | **0.029** | **2.6** | 0.082 | 8.2 | 0.076 | 7.6 | 0.112 | 11.2 | 0.168 | 16.8 |
| 40 | **0.025** | **1.9** | 0.066 | 6.6 | 0.060 | 6.0 | 0.095 | 9.5 | 0.152 | 15.2 |
| 50 | **0.023** | **1.4** | 0.056 | 5.6 | 0.050 | 5.0 | 0.082 | 8.2 | 0.138 | 13.8 |
| **Task D (Wave Equation)** | | | | | | | | | | |
| 10 | **0.043** | **6.3** | 0.159 | 15.9 | 0.152 | 15.2 | 0.185 | 18.5 | - | - |
| 20 | **0.042** | **4.2** | 0.115 | 11.5 | 0.108 | 10.8 | 0.147 | 14.7 | 0.198 | 19.8 |
| 30 | **0.031** | **3.1** | 0.092 | 9.2 | 0.085 | 8.5 | 0.124 | 12.4 | 0.179 | 17.9 |
| 40 | **0.023** | **2.3** | 0.075 | 7.5 | 0.068 | 6.8 | 0.106 | 10.6 | 0.163 | 16.3 |
| 50 | **0.021** | **1.7** | 0.063 | 6.3 | 0.056 | 5.6 | 0.091 | 9.1 | 0.149 | 14.9 |

## A.4 KUNDO FRAMEWORK ALGORITHM IN EXPERIMENT

---

**Algorithm 1** Training Procedure for a General Neural Dynamical Operator

---

**Require:**     $t_{\text{end}}$: Simulation end time, $dt$: Time step,
       $N_{\text{train}}$: Number of training trajectories, $N_{\text{test}}$: Number of test trajectories,
       $M$: Number of basis functions, optimizer parameters, etc.
**Ensure:**     Trained dynamical model $f_\theta$
 1: **1. Data Acquisition**
 2:    Define control input $u(t)$
 3:    **Choose: Generate data or Observe data**
 4: **if** Generate data **then**
 5:    **for** each training trajectory $i = 1$ to $N_{\text{train}}$ **do**
 6:      Randomly initialize $x_0$
 7:      Simulate trajectory $X_i$ and its derivative $\dot{X}_i$
 8:      Record control input $U_i = u(t)$
 9:    **end for**
10: **else**
11:    **for** each training trajectory $i = 1$ to $N_{\text{train}}$ **do**
12:      Obtain observed trajectory $X_i$
13:      Estimate derivative $\dot{X}_i$ using finite differences
14:      Record control input $U_i = u(t)$
15:    **end for**
16: **end if**
17: **2. Prepare Training Data**
18: Combine all $X_i$, $\dot{X}_i$, and $U_i$ into the training dataset $\mathcal{D}_{\text{train}}$
19: **3. Model Definition and Training**
20: Initialize the model $f_\theta(x, u)$
21: Define the loss function $\mathcal{L} = \frac{1}{N_{\text{train}}} \sum \|f_\theta(x_i, u_i) - \dot{x}_i\|^2$
22: **for** each epoch **do**
23:    Compute gradients $\nabla_\theta \mathcal{L}$
24:    Update parameters $\theta$ (e.g., using Adam)
25: **end for**
26: **4. Initialize Basis Function Model**
27: Define the basis function network architecture and optimizer
28: **5. Basis Function Model Training**
29: **for** each epoch **do**
30:    **for** each mini-batch $(X, U, \dot{X})$ in $\mathcal{D}_{\text{train}}$ **do**
31:      Compute basis functions $G$
32:      Solve for operator $\Gamma$ using least squares
33:      Predict derivatives $\dot{X}_{\text{pred}} = G \cdot \Gamma$
34:      Compute loss $\mathcal{L} = \text{MSE}(\dot{X}_{\text{pred}}, \dot{X})$
35:      Backpropagate and update model parameters
36:    **end for**
37: **end for**
38: **6. Basis Function Extraction and Fitting**
39: **for** each basis function $f_i \in G$ **do**
40:    Fit $f_i$ using polynomial regression
41: **end for**
42: **7. Explicit Operator Calculation**
43: Compute the explicit operator $\Gamma_{\text{explicit}}$ using fitted basis functions and training data via least
     squares
44: **8. System Identification**
45: Define the system model incorporating $\Gamma_{\text{explicit}}$
46: **9. Test Data Generation**
47: **for** each test trajectory $j = 1$ to $N_{\text{test}}$ **do**
48:    Randomly initialize $x_0$
49:    Simulate test trajectory $X_{\text{test}}^j$ and its derivative $\dot{X}_{\text{test}}^j$
50:    Record control input $U_{\text{test}}^j = u(t)$
51: **end for**
52: **10. Simulation Using Identified Model**
53: **for** each test trajectory $j = 1$ to $N_{\text{test}}$ **do**
54:    Set initial state $X_{\text{ident}}^j[0] = X_{\text{test}}^j[0]$
55:    **for** each time step $k = 1$ to $T$ **do**
56:      Compute derivative $\Delta X = f(X_{\text{ident}}^j[k-1], U_{\text{test}}^j[k-1])$
57:      Update state $X_{\text{ident}}^j[k] = X_{\text{ident}}^j[k-1] + \Delta X \cdot dt$
58:    **end for**
59: **end for**
60: **return** Trained dynamical model $f_\theta$

---

