# OpenReview forum: "Koopman Universal Neural Dynamic Operator: Achieving Fully Explicit Expression Identification for Nonlinear Dynamical Systems"
_ICLR.cc/2025/Conference — ICLR 2025 Conference Withdrawn Submission_

### Official Review · Reviewer_4gU7 · 2024-10-29

**Soundness:** 3
**Presentation:** 2
**Contribution:** 2
**Rating:** 3
**Confidence:** 3

**Summary:**

This paper proposes the Koopman Universal Neural Dynamic Operator (KUNDO) to identify fully explicit expressions of complex nonlinear systems. It combines the non-trivial concepts of neural networks, Koopman operator theory, and the universal approximation theorem to learn a set of Koopman-compatible basis functions. By encoding dynamical information through Koopman operator theory, the proposed framework achieves high prediction accuracy and even extrapolates well beyond the training regime.

**Strengths:**

1. The paper is well-written and well-organized, and comprehensive experiments are provided.
2. The architecture is novel, and a complete theoretical analysis is given.
3. It is less sensitive to noise and hyperparameters and extrapolates beyond the training regime.

**Weaknesses:**

The paper makes several claims that don't seem to be substantiated by the results provided in the paper. Please see the following comments for details.
1. Throughout the introduction and manuscript, a consistent claim made by the authors is that KUNDO can identify fully explicit expressions of complex nonlinear systems. However, this claim is not demonstrated in the manuscript, either in the form of expressions, stimuli, or feature maps. The authors should provide evidence in support of their claim on the interpretability.
2. It is also claimed that KUNDO demonstrates remarkable efficiency in small sample scenarios, but the numerical section uses 100 samples in each example, with each sample size ranging in the order of 1000. The authors should consider fewer samples and data points to illustrate their method's strength in small data scenarios.
3. Alongside the LSTM, NODE, and GPR, the authors should have considered more robust methods like neural operators [1-3] and Bayesian methods for equation discovery [4,5]. Neural operators also tend to learn generalizable features, and Bayesian methods provide good system identification with low data limits. Hence, it would be a good case study to see where the proposed KUNDO is placed among all the methods.
4. The paper claims that KUNDO provides unprecedented interpretability and analytical capabilities, especially for systems with higher dimensions and stronger nonlinearities. This seems to be vague to me. While chaotic Lorenz and the robotic arm examples are well appreciated, Task A is weakly nonlinear, and Task D is a linear PDE. A few complex nonlinear examples involving nonlinear 2D PDEs are required to substantiate such a claim.

[1] Li, Zongyi, et al. "Fourier neural operator for parametric partial differential equations." arXiv preprint arXiv:2010.08895 (2020).\
[2] Hao, Zhongkai, et al. "Gnot: A general neural operator transformer for operator learning." International Conference on Machine Learning. PMLR, 2023.\
[3] Lu, Lu, et al. "Learning nonlinear operators via DeepONet based on the universal approximation theorem of operators." Nature machine intelligence 3.3 (2021): 218-229.\
[4] Nayek, Rajdip, et al. "On  spike-and-slab priors for Bayesian equation discovery of nonlinear  dynamical systems via sparse linear regression." Mechanical Systems and Signal Processing 161 (2021): 107986.\
[5] Bonneville, Christophe, and  Christopher Earls. "Bayesian deep learning for partial differential  equation parameter discovery with sparse and noisy data." Journal of Computational Physics: X 16 (2022): 100115.

**Questions:**

Please see below questions on the paper content:
1. line 151. The statement on the systematic framework needs to be clarified. Why do the authors feel that physics-informed, physics-encoded, and other differentiable physics-based frameworks do not provide systematic integration?
2. Eq. (6). has typos.
3. line 221. "$\Phi(x, u)$ is a vector composed of polynomial-approximated basis functions." But it is estimated using NN, right?
4. Fig. (5). It would be better if the results were tabulated in a table and a relative percentage error was considered for better comparison.
5. line 324. Is it not supposed to be 100 points since $\Delta x$ = 0.1?
6. I think this paper will benefit from a convergence study of a number of basis functions.
7. Please mention the size of the model parameters and computational time for each method.

---

### Official Review · Reviewer_cPo5 · 2024-10-31

**Soundness:** 3
**Presentation:** 3
**Contribution:** 3
**Rating:** 6
**Confidence:** 3

**Summary:**

The submission “Koopman Universal Neural Dynamic Operator (KUNDO): Achieving Fully Explicit Expression Identification for Nonlinear Dynamical Systems" presents a new framework called KUNDO for modeling and identifying complex nonlinear dynamical systems. As far as I understand, the main objective of KUNDO is to bridge the gap between conventional mathematical modeling and machine learning approaches by giving out fully explicit mathematical expressions of underlying dynamics for the system of interest while maintaining high predictive accuracy.

The key contributions of this submission are
a. A method for the integration of neural networks with Koopman operators to learn Koopman-compatible basis functions.
b. Full and explicit expression identification since the claim is that the KUNDO model transforms strongly nonlinear dynamics into fully explicit mathematical forms, providing interpretable models of complex systems. The claim is that by representing the system dynamics using a linear mapping in a higher-dimensional space, the method yields explicit expressions that describe the system's behavior.
c. Enhanced interpretability of the model by reduction of the reliance on manually selected basis functions, allowing the neural network to learn appropriate functions automatically.
d. Finally, KUNDO demonstrates strong performance even with limited data and
All the above are accompanied by some good theoretical analysis as well as somewhat broad validation on diverse nonlinear systems.

**Strengths:**

I believe that this submission presents indeed a novel contribution that integrates neural networks with Koopman operator theory to achieve this fully explicit expression identification which to my best understanding is in general hard to achieve. In terms of originality, the paper stands out to me exactly because of the reduction of the need the rely on manually selected observables. This approach both, as it seems, improves the modeling of nonlinear systems but also enhances interpretability thereof by providing explicit mathematical expressions. Furthermore, the paper demonstrates rigorous theoretical development and somewhat comprehensive experimental validation. The authors provide a limited mathematical analysis of KUNDO's properties, including proofs of convergence, stability, generalization ability, and robustness. (That said, on a first read it was not clear that the details are in the Appendix. I would like to see something e.g. about convergence as a Theorem in the main body). The experiments cover a wide range of systems—from classical nonlinear systems like the Lorenz system to real-world data from a robotic arm and a one-dimensional wave equation. The comparison with several baseline methods shows that KUNDO consistently outperforms them in various metrics, supporting the central claims of the paper. The clarity of the paper is good, indeed. It is well-organized and guides the reader through the complex interplay between neural networks and Koopman operator theory. The authors provide sufficient context and explanations to make the content accessible to readers with a background in dynamical systems and machine learning. The reference list is good.
In terms of significance, lacking some domain expertise on the practicalities, the paper seems to address an important problem in modeling complex nonlinear dynamical systems, which has implications across various scientific and engineering domains The demonstrated efficiency of KUNDO in small sample scenarios and its robustness to noise enhance my suspicions on the potential of its practical applicability. Overall, the paper contributes somewhat valuable insights and tools to the broader ICLR community which justifies my scores.

**Weaknesses:**

Despite the many strengths identified on the paper, there are few weaknesses that could be addressed.
I find that the paper has insufficient experimental details. While the paper presents experimental results demonstrating the effectiveness of KUNDO, it lacks detailed descriptions of the experimental setup, which hampers reproducibility and thorough assessment.

More concretely, the paper mentions using a neural network to learn Koopman-compatible basis functions but provides minimal information about the network architecture. See below Eq. (1): “This network can encompass various architectures, such
as feedforward neural networks, CNNs, etc [...] or combinations thereof.” Ideally, some details such as the number of layers, types of layers (e.g., fully connected, convolutional), activation functions used, and any normalization techniques are missing. This information is crucial for others to replicate the experiments and understand the role of the architecture in the model's performance. Secs 4.1.1 to 4.1.4 maybe could contain this type of information? Also, there is absence of  information on the hyperparameters used in training KUNDO and the baseline models. Anything from learning rates, batch sizes, regularization parameters, optimizer types,etc. The paper also does not detail the training procedures, such as the number of epochs, early stopping criteria, or how overfitting was prevented. Similarly, the evaluation metrics, while mentioned, lack precise definitions, and it is not clear whether standard deviations are computed over multiple runs or across different trajectories. Discussion of the above would be beneficial for the practitioners who otherwise seem that would be in a jungle of possibilities for their domain application.

**Questions:**

One of the key claims of the paper is that KUNDO enhances interpretability by providing fully explicit mathematical expressions of system dynamics. However, the paper does not provide concrete examples or case studies illustrating how these explicit expressions lead to intuitive insights or deeper understanding of the systems studied. Can the authors address this?

I would appreciate if the authors could highlight scenarios where KUNDO's explicit expressions provide advantages over the so-called traditional methods (name which ones?) or black-box machine learning models. Can the authors please address this?

For the experiments can the authors provide quantitative measures of training time, memory usage, and computational resources (e.g., compute hours) which would give readers a clearer understanding of the practical feasibility of KUNDO?

Can the authors provide more info on the scaling of KUNDO and instance types where they believe the model would perform below par? Have they found such cases? If so, what happens next? Is there a path to address this?

In relation to the above, are classes of systems where KUNDO is known to not perform well? If so why?

---

### Official Review · Reviewer_tTnr · 2024-11-02

**Soundness:** 2
**Presentation:** 2
**Contribution:** 1
**Rating:** 3
**Confidence:** 4

**Summary:**

This paper proposes KUNDO, a Koopman Universal Neural Dynamics Operator by integrating Koopman operator with neural network. The Koopman coordinates identified are approximated using polynomial basis and the final equation is represented in terms of the same. Numerical examples (mostly simple) are presented to illustrate the applicability of the method.

**Strengths:**

The paper is mostly well written and the idea of expressing Koopman coordinate using basis function is new.

**Weaknesses:**

1) For a practical implementation point of view, there is supposed to be a large number of basis functions to approximate the theoretically infinite dimensional Koopman coordinates. Each basis function is further approximated using polynomial basis. So overall, there are large number of basis functions present in the final model rendering the system non-interpretable (which is opposite of one of the central claim of the paper).
2) With this approach, the final equation obtained is only in terms of basis functions (potentially polynomial or Fourier). However, real-life systems exhibit spatio-temporal behavior and modeled using partial differential equations. It is not clear why (and how) the method should generalize in such cases.
3) The central idea behind interpretable model is to have a model (potentially sparse) where each term conveys a specific meaning. Just having explicit expression does not make a model interpretable. For example, each neuron in MLP has an explicit form g(wx+b) and the final network is just composition of such functions. This does not make MLP interpretable.
4) The numerical examples presented are quite simple with no spatio-temporal system. Including spatio-temporal system (E.g., Navier Stokes equation).
5) The method require 1000s of trajectories while its SINDy (its competitor) require a single realization. There is no evidence in the manuscript that KUNDO can learn from a single trajectory.
6) Generalization in physics means the learned model should work with different forcing functions. However, the proposed approach only considers initial condition driven systems with no forcing function.

**Questions:**

1) The primary contribution of this work resides in identification of basis function using Koopman Operator. However, we know that Koopman Coordinates are infinite dimensional, and truncation therein results in error. How do you ensure that the basis function (Koopman coordinates) learnt using the proposed approach is adequate in representing the dynamics? Is it based on validation data?
2) As mentioned in the weakness, will the method work with a single realization (as in case of SINDy)?
3) Following up on point 2) in the weakness section, it is not clear why and how the model should work for PDE? Clearly, Eq. 7 does not contain any spatial derivative. So, what is KUNDO learning here? Is it learning coupled ODEs? If yes, then is the dimensionality of the learned ODE dependent on the spatial discretization? If the answer is yes, then that is a major drawback as the learned model will fail to capture in future evolution at non-grid points, rendering the model ineffective for practical use.
3) As the formulation includes u (control force), it is better to include an example that has forcing term. A simple example in this context can be the forced duffing oscillator. Using this example, it can also be investigated if the method generalizes to new force (say u in training data is from sin(wt)) and during prediction it is some realization of a white noise. SINDy seamlessly generalizes to these scenarios and hence, it is important illustrate the same here.
4) How the method performs with change in nonlinearity of the system. SINDy often fails to identify the nonlinear term when the nonlinearity is mild. Is the proposed approach immune to this?

---

### Official Review · Reviewer_GZg4 · 2024-11-04

**Soundness:** 2
**Presentation:** 1
**Contribution:** 1
**Rating:** 1
**Confidence:** 4

**Summary:**

This paper introduces KUNDO (Koopman Universal Neural Dynamic Operator), an approach that learns Koopman compatible basis functions using neural networks. KUNDO then uses these learned basis functions within the EDMD framework to get the operator approximation. Hence KUNDO presents a hybrid method that is able to achieve system identification for complex non-linear systems while providing models with fully explicit mathematical expressions.
The method is able to deal with noise in the input and small sample size availability for the learning task.

**Strengths:**

The authors through their paper have tried to address an important problem in the realm of system identification using hybrid models. The proposed approach exploits the functional approximation ability of Neural networks with Koopman operator theory.

The proposed approach seems to work well even when data is not abundantly available.

**Weaknesses:**

I find the paper lacking in terms of recent literature and methods. This makes it challenging to identify the novelty of the paper. The paper requires significant revisions.
1. Lacking ablation study : The authors suggest use of Fourier series as basis functions but didn't present any experimental results on them. I would also like to encourage the authors to present extra results regarding the spectral analysis of the learned Koopman operator especially in the case of small sample size learning experiment.
2. Lacking novelty : The paper severely lacks novelty in the light of some of the recent works listed below. Several works have shown the effectiveness of learning Koopman operator and the basis functions using neural networks. Please look at the list of references below.
3. Lacking crucial comparisons : None of the baselines presented in the paper are among the recent works or represent state of the art. Hence a thorough literature review and detailed comparison is crucial.
4. Missing crucial references : The authors have not cited several crucial works in this area. A quick look at the reference suggests no reference post 2021 in the reference section. Several works have been proposed in this area that are crucial to be discussed and compared with,

Inzerilli, Prune, et al. "Consistent long-term forecasting of ergodic dynamical systems." arXiv preprint arXiv:2312.13426 (2023).

Kostic, Vladimir, et al. "Learning dynamical systems via Koopman operator regression in reproducing kernel Hilbert spaces." Advances in Neural Information Processing Systems 35 (2022): 4017-4031.

Wang, R., et al. "Koopman Neural Forecaster for Time Series with Temporal Distribution Shifts." International Conference on Learning Representations (ICLR). 2023.

Lew, Ethan, et al. "Autokoopman: A toolbox for automated system identification via koopman operator linearization." International Symposium on Automated Technology for Verification and Analysis. Cham: Springer Nature Switzerland, 2023.

Wilson, Dan. "Koopman operator inspired nonlinear system identification." SIAM Journal on Applied Dynamical Systems 22.2 (2023): 1445-1471.

Rostamijavanani, Abdolvahhab, Shanwu Li, and Yongchao Yang. "A study on data-driven identification and representation of nonlinear dynamical systems with a physics-integrated deep learning approach: Koopman operators and nonlinear normal modes." Communications in Nonlinear Science and Numerical Simulation 123 (2023): 107278.

Sun, Zexin, Mingyu Chen, and John Baillieul. "Koopman-based Deep Learning for Nonlinear System Estimation." arXiv preprint arXiv:2405.00627 (2024).

**Questions:**

Please look at the weakness section.

---

### Note · Authors · 2024-11-13

I have read and agree with the venue's withdrawal policy on behalf of myself and my co-authors.